# Androgen receptor signalling in macrophages promotes TREM-1-mediated prostate cancer cell line migration and invasion

Bianca Cioni [1,10], Anniek Zaalberg [1,10], Judy R. van Beijnum [2], Monique H. M. Melis[3], Johan van Burgsteden[3], Mauro J. Muraro[4], Erik Hooijberg [5], Dennis Peters[6], Ingrid Hofland[6], Yoni Lubeck[5], Jeroen de Jong[5], Joyce Sanders [5], Judith Vivié[4], Henk G. van der Poel[7], Jan Paul de Boer[7], Arjan W. Griffioen[2], Wilbert Zwart [1,8,9 ✉] & Andries M. Bergman [1,7 ✉]

The androgen receptor (AR) is the master regulator of prostate cancer (PCa) development, and inhibition of AR signalling is the most effective PCa treatment. AR is expressed in PCa cells and also in the PCa-associated stroma, including infiltrating macrophages. Macrophages have a decisive function in PCa initiation and progression, but the role of AR in macrophages remains largely unexplored. Here, we show that AR signalling in the macrophage-like THP-1 cell line supports PCa cell line migration and invasion in culture via increased Triggering Receptor Expressed on Myeloid cells-1 (TREM-1) signalling and expression of its downstream cytokines. Moreover, AR signalling in THP-1 and monocyte-derived macrophages upregulates IL-10 and markers of tissue residency. In conclusion, our data suggest that AR signalling in macrophages may support PCa invasiveness, and blocking this process may constitute one mechanism of anti-androgen therapy.

[1] Divisions of Oncogenomics, The Netherlands Cancer Institute (NKI), Plesmanlaan 121, 1066CX Amsterdam, The Netherlands. [2] Angiogenesis laboratory, Medical Oncology, Amsterdam UMC, Cancer Center Amsterdam, De Boelelaan 1117, 1081 HV Amsterdam, The Netherlands. [3] Molecular Genetics, NKI, Plesmanlaan 121, 1066CX Amsterdam, The Netherlands. [4] Hubrecht Institute - KNAW and University Medical Center Utrecht, Uppsalalaan 8, 3584CT Utrecht, The Netherlands. [5] Division of Pathology, NKI, Plesmanlaan 121, 1066CX Amsterdam, The Netherlands. [6] Core Facility Molecular Pathology, NKI, Plesmanlaan 121, 1066CX Amsterdam, The Netherlands. [7] Urology and Medical Oncology, NKI, Plesmanlaan 121, 1066CX Amsterdam, The Netherlands. [8] Laboratory of Chemical Biology and Institute for Complex Molecular Systems, Department of Biomedical Engineering, Eindhoven University of Technology, PO Box 513, 5600MB Eindhoven, The Netherlands. [9] Oncode Institute, The Netherlands. [10] These authors contributed equally: Bianca Cioni, Anniek Zaalberg. ✉email: w.zwart@nki.nl; a.bergman@nki.nl

Prostate cancer (PCa) is the second most-common cancer in men worldwide and accounts for 300.000 deaths annually[1]. During normal development of the prostate, epithelial–stromal interactions help maintaining the physiological homoeostasis of the prostate[2]. However during PCa development, stromal cells can change in phenotype to support tumour progression instead[3]. This 'reactive stroma' is composed of many non-immune and immune cells including fibroblasts, endothelial cells, lymphocytes and macrophages, which can support PCa progression predominantly by secreting soluble factors into the extra cellular matrix[4].

Macrophages are antigen presenting cells (APCs) that are derived from embryonic precursors and circulating CD14+ monocytes originating from the bone marrow[5]. A large spectrum of tumour-associated macrophage (TAM) phenotypes has been described, ranging from classically activated, pro-inflammatory and anti-tumour M1, to the anti-inflammatory and pro-tumour M2 macrophages[6,7]. TAM infiltration into PCa was associated with disease progression after hormonal therapy and preclinical studies suggested that TAMs support PCa cell proliferation and migration[8–10].

Physiological maintenance of the prostate strongly depends on androgen receptor (AR) signalling, which is also crucial for PCa development. While a large number of studies addressed the role of AR in PCa cells, only few reports focused on the molecular mechanisms of AR in stromal cells and its consequences for PCa progression and treatment in trans[11,12]. Expression of AR in macrophages was established in mice; however, the functionality of AR signalling in macrophages in relation to cancer development remained largely unknown[9,13,14].

In this study, we provide gene regulation data on AR signalling in human macrophages and show that activation of AR signalling in macrophages increases migration and invasion of PCa-derived cancer cells, mediated by upregulation of the Triggering Receptor Expressed on Myeloid cells-1 (TREM-1) receptor and its downstream cytokines and promotion of TAM differentiation. Our study illustrates that AR signalling in macrophages might represent a druggable cascade in the treatment of PCa patients.

## Results

**PCa-associated macrophages express the AR.** Even though AR is predominantly expressed in prostate epithelial cells, this receptor is also expressed in stromal cells. To establish AR expression in macrophages at the protein level, formalin-fixed paraffin embedded (FFPE) prostatectomy specimen of untreated PCa patients were stained for AR and CD163, a marker of tissue-resident macrophages including TAMs[15]. Figure 1b shows double staining of AR and CD163 in the PCa-associated stroma, suggesting AR expression in TAMs at the protein level. Multiplex immunofluorescence staining was performed to quantify AR in cells expressing CD163, and/or the myeloid cell markers HLA-DRA and CD14 in FFPE prostatectomy specimens of 20 patients. AMACR staining was used to annotate the tumour area (Fig. 1b), the 200 μm tumour border zone and distant normal prostate tissue. Expression of AR, CD163, HLA-DRA and CD14 was quantified in all three areas (Fig. 1c). AR was expressed in a median of 32.9% of CD163 and/or HLA-DRA and/or CD14 expressing cells in the Tumour area, which was not significantly different from cells in the tumour border or in the distant area (median 34.2% and 35.2%, respectively) (Fig. 1d).

In conclusion, these results suggest that myeloid cells, including macrophages infiltrating the PCa-associated stroma, express AR.

**AR is functional in macrophages.** Our results suggest that AR is expressed in macrophages that infiltrate into the PCa-associated stroma. As a working model to study AR functions in macrophages, monocytic THP-1 cells were PMA-activated in vitro into CD68+ macrophages (THP-1^PMA), as previously described (Fig. 2a)[16]. THP-1^PMA cells were further differentiated into classically activated macrophage-like cells by IFN-γ and LPS (THP-1^PMA;IFNG;LPS). In THP-1^PMA;IFNG;LPS cells, AR was expressed at the RNA and protein level (Fig. 2b, c, respectively). M14 melanoma cells were included as a negative control and did not express AR. Nuclear translocation and subsequent chromatin binding upon testosterone stimulation are crucial for AR to exert its transcription factor function[17]. Subcellular fractionation of THP-1^PMA;IFNG;LPS cells showed enrichment of AR in the chromatin fraction upon stimulation with the testosterone analogue R1881 compared to vehicle control (DMSO), suggesting translocation and chromatin interactions of AR in macrophage-like cells (Fig. 2d).

Next, these findings were verified in monocyte-derived macrophages (MDMs) (Supplementary Fig. 1). CD14+ cells were isolated from buffy coats of healthy male donors' blood and stimulated with GM-CSF, IFN-γ and LPS to promote cell maturation into MDMs (Supplementary Fig. 1A). As shown in Supplementary Fig. 1B, AR was readily expressed at the mRNA level in MDMs of three different donors. AR nuclear translocation was evaluated in MDMs that were stimulated with vehicle control or R1881 after hormone-deprivation. MDMs were stained for AR and CD68 as a pan-macrophage marker. AR was found both in the cytoplasm and in the nucleus of MDMs in vehicle-treated cells, while AR concentrated in the nucleus upon R1881 stimulation (Supplementary Fig. 1C), suggesting AR translocation and chromatin binding upon testosterone stimulation in MDMs.

These data suggest that AR is expressed and functional in human macrophages.

**AR signalling in THP-1 affects PCa cell migration.** To explore the potential consequences of AR signalling in macrophages on PCa proliferation and migration in trans, the CWR-R1 PCa-derived cancer cell line was exposed to conditioned medium (CM) of AR-activated THP-1^PMA;IFNG;LPS cells, to assess proliferation and migration. THP-1^PMA;IFNG;LPS cells were cultured in testosterone-containing, foetal bovine serum (FBS) proficient medium to activate AR signalling, while AR signalling was blocked by adding the AR signalling inhibitor RD162 to the culture medium (Fig. 3a). After 24 h, medium was removed and cells were extensively washed and subsequently cultured in hormone-deprived dextran-coated charcoal stripped (DCC) FBS proficient medium (DCC medium) for 48 h, after which the CM was harvested for downstream analyses. Interestingly, CM from vehicle-treated THP-1^PMA;IFNG;LPS cells significantly enhanced migration of CWR-R1 PCa cells compared to normal medium, while migration of CWR-R1 PCa cells was significantly reduced when cultured in CM of RD162-treated THP-1^PMA;IFNG;LPS cells compared to CM of vehicle-treated THP-1^PMA;IFNG;LPS cells (Fig. 3b, quantified in Fig. 3c). As expected, no migration of CWR-R1 cells was observed when cultured in hormone-deprived DCC medium. CWR-R1 proliferation was not affected by THP-1^PMA;IFNG;LPS CM (Supplementary Fig. 2), suggesting that AR signalling in macrophage-like cells regulates migration but not proliferation of PCa cells in vitro. To further validate our findings in a second PCa cell line, migration assays were repeated with PC3 cells. As PC3 cells do not express AR, this experimental setup enables us to exclude any potential carry-over of drug in the CM, which might affect AR signalling. As shown in Supplementary Fig. 3A and B, reduced migration of PC3 cells was observed when

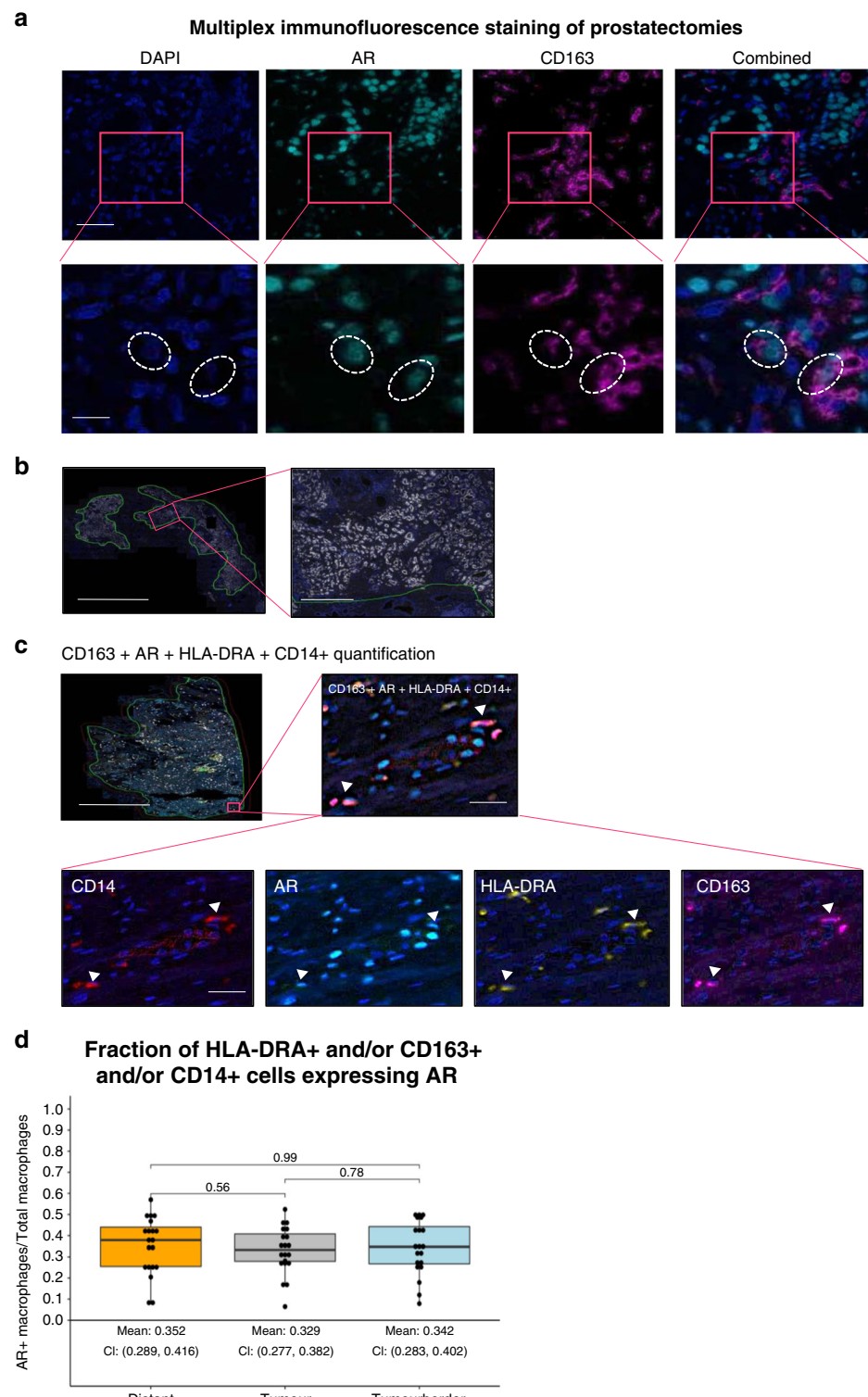

**a** Multiplex immunofluorescence staining of prostatectomies

DAPI  AR  CD163  Combined

**b**

**c** CD163 + AR + HLA-DRA + CD14+ quantification

CD163 + AR + HLA-DRA + CD14+

CD14  AR  HLA-DRA  CD163

**d** Fraction of HLA-DRA+ and/or CD163+ and/or CD14+ cells expressing AR

AR+ macrophages/Total macrophages

0.99
0.56
0.78

Mean: 0.352
CI: (0.289, 0.416)
Distant

Mean: 0.329
CI: (0.277, 0.382)
Tumour

Mean: 0.342
CI: (0.283, 0.402)
Tumourborder

cultured in CM from THP-1[PMA;IFNG;LPS] cells exposed to RD162, fully confirming our previous results.

In conclusion, these findings suggest that AR signalling in macrophages results in the secretion of factors that promote PCa cell migration.

**AR inhibition in THP-1 cells suppresses tumour growth**. To assess the consequences of AR signalling in macrophages for the in vivo anchorage independent growth and systemic dissemination of PCa cells, we used the chick embryo chorioallantoic membrane (CAM) assay. On embryonic development day (EDD)6, PC3 and LNCaP PCa cells were inoculated into the CAM. The cells successfully engrafted formed tumours that grew over time. From EDD10 onwards, eggs were treated with either DMSO-treated THP-1[PMA;IFNG;LPS] CM, RD162-treated THP-1[PMA;IFNG;LPS] CM, CCL2 cytokine as a positive control or NaCl as a negative control. Treatment was applied every day until EDD14 and tumours were harvested at EDD17. As shown in Fig. 4a and quantified in Fig. 4b, PC3 tumour growth was significantly reduced in eggs treated with RD162 CM compared to

**Fig. 1 AR expression in PCa-resident macrophages. a** Immunofluorescence staining of a FFPE prostatectomy specimen from a systemically untreated PCa patient showing the presence of AR in CD163+ cells. Nuclei were stained with DAPI (dark blue), whereas AR and CD163 were visualized in light blue and purple, respectively (scale bar = 100 μm). Lower panel are magnifications of inserts (scale bar = 50 μm). Dotted circles identify DAPI+, AR+ and CD163+ cells. These images are representative of immunofluorescence stainings performed in FFPE prostatectomy specimen from three different patients. Pictures were taken in at least five areas to assess marker expression. **b** Multiplex immunofluorescence analysis. AMACR staining indicating the tumorous area. Representative image of 200–300 scans. Scale bar = 5000 μm (Left panel), 500 μm (Right panel; insert). **c** Multiplex immunofluorescence analysis. Representative tumorous area in a FFPE prostatectomy specimen stained for CD163, AR, HLA-DRA and CD14 and all combined. Each triangle represents a positive cell included in the quantification. Representative image of 200–300 scans. Scale bar = 5000 μm (Top left panel), scale bar = 80 μm (inserts). **d** Quantification of multiplex immunofluorescence analysis. Boxplot (median values with interquartile range) showing fraction of HLA-DR+ and/or CD163+ and/or CD14+ cells expressing AR, in the tumour area, in the 200 μm tumour border zone around the tumour area and in the area distant from the tumour in 20 FFPE prostatectomy specimen. Datapoints show individual patients. *p*-Values were calculated using a Wilcoxon rank-sum test with a cutoff for significance of 0.05. Source data are provided as a source datafile.

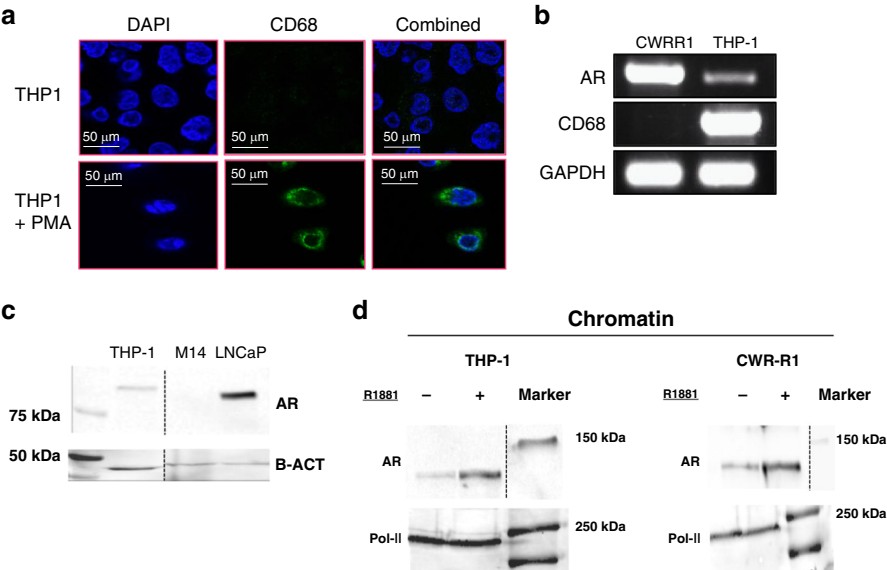

**Fig. 2 AR expression and nuclear translocation in THP-1 cells. a** Expression of the macrophage marker CD68 (green) in THP-1 cells at steady state (no treatment) and following 2 days of PMA stimulation. Nuclei were stained with DAPI (dark blue). Scale bar = 50 μm. Representative of three images per condition. **b** RT-PCR showing *AR* expression at the RNA level in human cancer cell lines of prostate epithelial (CWR-R1) and monocytic (THP-1$^{PMA;IFNG;LPS}$) origin. *GAPDH* was used as a house-keeping control gene. This experiment was performed two times. **c** Western blot showing AR expression at the protein level in human cell lines originated from prostate cancer (LNCaP), melanoma (M14) and monocytic leukaemia (THP-1$^{PMA;IFNG;LPS}$). β-Actin was used as a loading control. This experiment was performed two times. Source data are provided as a source datafile. **d** Western blot showing AR expression at the protein level in the subcellular chromatin fraction of THP-1$^{PMA;IFNG;LPS}$ cells and CWR-R1 human PCa cells upon R1881 stimulation. Pol-II was used as a loading control of the chromatin fraction. This experiment was performed two times. Source data are provided as a source datafile.

DMSO CM. As expected, treatment with CCL2 cytokine strongly increased tumour growth compared to NaCl control. Comparable, but no significant differences were obtained with LNCaP PCa cell grafts (Supplementary Fig. 4). These results suggest that androgen stimulated macrophage-like cells excrete soluble factors that promote anchorage independent growth of prostate tumours grafted into the CAMs. This is in contrast to our in vitro data, where there was no difference in 2D cell proliferation (Supplementary Fig. 2). To assess epithelial–mesenchymal transition (EMT) potential of PCa cells, the expression of the EMT marker human *vimentin* in CAM tissue distant from the implanted tumour (normal CAM) was evaluated. Expression of human *vimentin* in normal CAM suggested the presence of disseminated cells in all treatment conditions (Fig. 4c). However, there was no difference in the expression of *vimentin* in normal CAM between RD162 CM treated tumours and DMSO CM treated tumours.

In conclusion, we showed that AR stimulation of macrophage-like cells can promote anchorage independent growth of PCa cells in vivo, but not EMT.

**AR regulates the TREM-1 pathway in macrophages**. How does AR activation in macrophages stimulate PCa cell migration? To address this question, we analyzed the genome-wide chromatin binding profiles of AR in THP-1$^{PMA;IFNG;LPS}$ cells, using chromatin immunoprecipitation followed by massive parallel DNA sequencing (ChIP-seq). THP-1$^{PMA;IFNG;LPS}$ cells were stimulated with R1881 or vehicle (DMSO) for 4 h prior to ChIP-seq. As AR is an enhancer-selective transcription factor[18], ChIP-seq for the enhancer-selective histone modification H3K27ac was included. As expected, based on the AR chromatin fractionation analyses (Fig. 2d), R1881 stimulation strongly increased the number of AR-specific binding sites in THP-1$^{PMA;IFNG;LPS}$ cells (vehicle: 97 sites, R1881: 5072 sites), while H3K27Ac was not affected by the hormone (Fig. 5a). Representative regions of both AR and H3K27ac peaks are shown in Fig. 5b. AR-binding sites were predominantly found at enhancer elements, including intronic and distal intergenic regions (Fig. 5c). AR and H3K27ac binding sites are represented in the heatmap in Fig. 5d. Motifs for members of the AP-1 complex, including FOS and JUN were found to be strongly enriched (Fig. 5e). Importantly, well-known

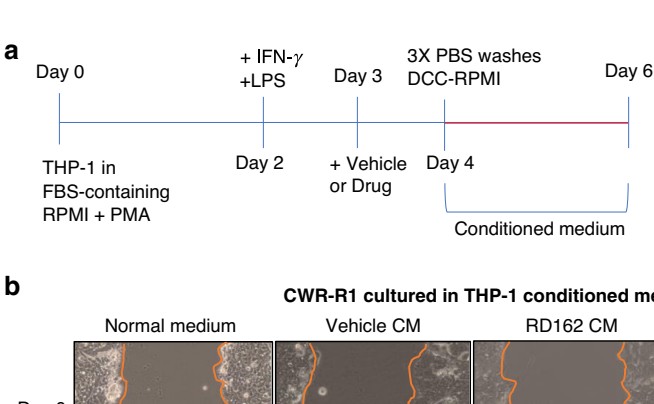

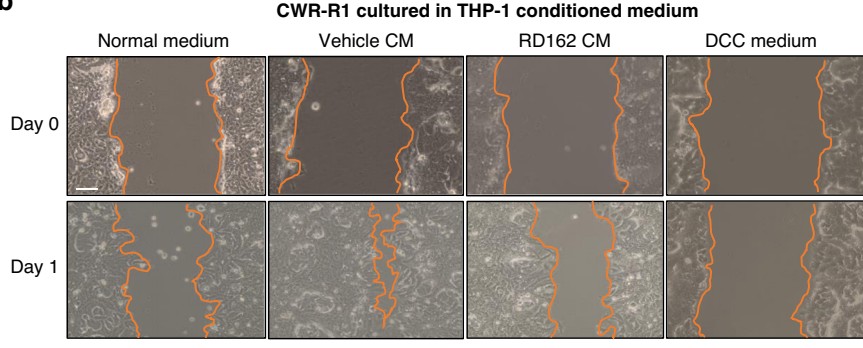

**b** CWR-R1 cultured in THP-1 conditioned medium

Normal medium    Vehicle CM    RD162 CM    DCC medium

Day 0

Day 1

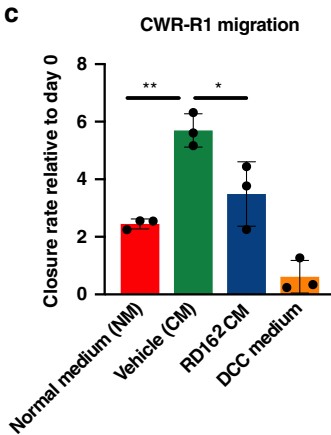

**c** CWR-R1 migration

**Fig. 3 AR signalling in THP-1 cells affects PCa cell line migration. a** Workflow showing the procedure of THP-1 cell differentiation into macrophage-like cells and generation of CM. THP-1 cells were stimulated with PMA (Day 0), IFN-γ and LPS (Day 2) and exposed to Vehicle or Drug on Day 3. After 24 h of stimulation, cells were carefully washed and replenished with fresh medium. Medium was collected as 'CM' after 48 h. **b** Scratch assay of CWR-R1 human PCa cells cultured in normal medium (NM) alone or in combination with CM of THP-1 cells stimulated with vehicle or RD162 at base line (0 h) and after 24 h. Charcoal-stripped (DCC) medium was used as a control. Representative of three different images per condition. Scale bar = 200 μm. **c** Quantification of three independent scratch assays. Datapoints show mean value of three technical replicates in each experiment. CWR-R1 cell migration is assessed through closure of the scratch after 24 h relative to day 0. Error bars represent the s.e.m.. *: $p = 0.02$ and **: $p = 0.002$. $p$-Values were calculated from means of three biological replicates using a One-way Anova test with a cutoff for significance of 0.05. Source data are provided as a source datafile.

transcription motifs involved in PCa physiology and pathology were not enriched at AR sites, including Androgen Response Elements (AREs)[19] and the well-known AR pioneer factor motifs GATA3, FOXA1 and HOXB13[20]. These results suggest that AR in THP-1[PMA;IFNG;LPS] cells binds the DNA via the AP-1 complex of co-factors, as we[12] and others[21,22] previously also observed for AR in fibroblasts.

These findings were confirmed in single-cell RNA-sequencing data of CD14+ and/or CD11b+ cells isolated from PCa biopsies, which were collected directly after prostatectomy and processed for single-cell mRNA sequencing (Supplementary Fig. 5A). The Sorting strategy is described in Supplementary Fig. 5B. HLA-II (*HLA-DR*+ and *HLA-DP*+) a marker of APCs, including macrophages and dendritic cells, was expressed at the single-cell level (Supplementary Fig. 6A). In these cells, no expression of the PCa epithelial cell marker Alpha-methylacyl-CoA racemase (*AMACR*), epithelial cell adhesion molecule (*EPCAM*) and the

mesenchymal cell marker Platelet-Derived Growth Factor Receptor β (*PDGFR-β*) was found, confirming their non-epithelial and non-mesenchymal origin. A small population of CD14+ and/or CD11b+ cells express Granzyme A (*GZMA*) and are likely natural-killer-like cells (Supplementary Fig. 6B). The majority of cells express the 'general' macrophage markers *CD68* and *CSFR1*, suggesting that these cells are macrophages (Supplementary Fig. 6B). Moreover, CD14+ and/or CD11b+ cells expressed the M2-like markers CD206, CD163 and IL-10, while expression of the M1-like markers STAT1, IL-12, CD80 and CXCL10 was generally low (Supplementary Fig. 6C and D). In the CD14+ and/or CD11b+ cells no FOXA1, HOXB13 and GATA3 expression was found, while members of the AP-1 complex including FOS, JUN and ATF were readily expressed (Fig. 5f).

To identify potential direct AR-target genes in THP-1[PMA;IFNG;LPS] cells, we identified all genes with an AR site <20 kb from the transcription start site or within the gene body. Ingenuity Pathway

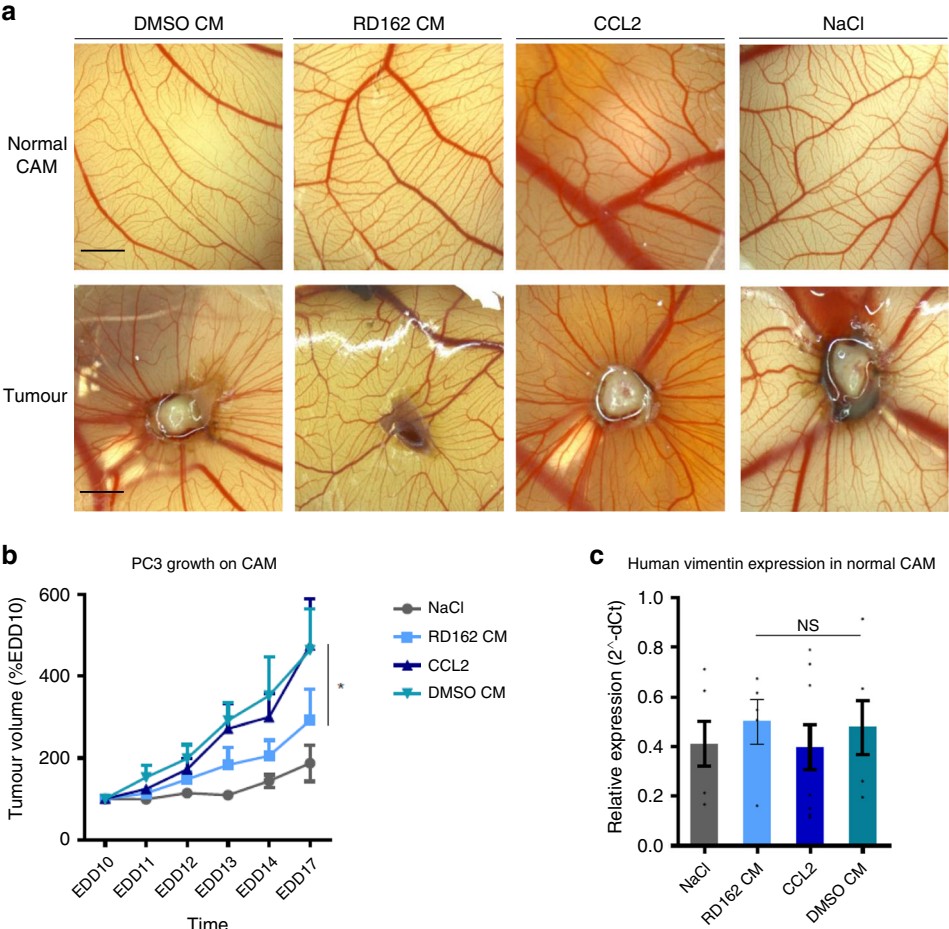

**Fig. 4 AR signalling in THP-1 cells affects prostate tumour growth in vivo. a** Representative images of both normal chick embryo chorioallantoic membrane (CAM) and PC3 PCa cells derived tumours growing on the CAMs replenished with CM of DMSO or RD162-treated THP-1[PMA;IFNG;LPS] cells, CCL2 or NaCl,. The only tumour grown in NaCl-treated CAM conditions is shown. All other NaCl-treated CAMs showed no sign of tumour growth. Scale bar = 500 μm. **b** Growth curves of PC3 tumours grafted into CAMs showing tumour volume over time in the different treatment conditions. Datapoints represent the average tumour volume as a percentage of the tumour volume at the start of treatment (EDD10). Error bars represent s.e.m. of 6–10 biological replicates per condition in one experiment with one batch of THP-1[PMA;IFNG;LPS] cell CM (DMSO CM and RD162 CM). *: $p = 0.02$. $p$-value comparing DMSO CM versus RD162 CM was calculated using a Two-way Anova with a cutoff for significance of 0.05. Source data are provided as a source datafile. **c** The expression of human *vimentin* in disseminated PC3 cells into CAM tissue distant from the primary tumour site (normal CAM) in the different treatment conditions and normalized to the reference gene human *Cyclophylin A* ($2^{-dCt}$). Error bars represent the s.e.m. of 5–9 biological replicates per condition in one experiment with one batch of THP-1[PMA;IFNG;LPS] cells CM (DMSO CM and RD162 CM). $p = 1.0$. $p$-value comparing DMSO CM versus RD162 CM was calculated using a One-way Anova test with a cutoff for significance of 0.05. NS: no significant difference. Source data are provided as a source datafile.

Analysis (IPA) on the resulting gene list revealed the TREM-1 signalling pathway as the most-enriched biological process regulated by AR in THP-1[PMA;IFNG;LPS] cells (Fig. 5g).

To validate our findings in primary cells, we performed AR and H3K27ac ChIP-seq in MDMs. Highly analogous to the THP-1[PMA;IFNG;LPS] data, AR binding in MDMs was strongly increased by R1881 stimulation compared to non-stimulated cells (Supplementary Fig. 7A and B. Vehicle: 512 sites, R1881: 9645 sites). AR-binding sites in MDMs were predominantly found in intronic and distal intergenic regions (Supplementary Fig. 7C) and ~50% of AR sites coincided with H3K27ac sites, which confirmed our observations in THP-1[PMA;IFNG;LPS] cells (Supplementary Fig. 7D). In full concordance with the observations in THP-1[PMA;IFNG;LPS] cells, AP-1 motifs were most-prominently found enriched at AR sites in MDMs, again confirming the results obtained in THP-1[PMA;IFNG;LPS] cells (Supplementary Fig. 7E). Importantly, also in MDMs, the TREM-1 signalling pathway was the most significant

biological pathway related to AR signalling (Supplementary Fig. 7F), further supporting our original observations that the TREM-1 signalling pathway is under the control of AR activity in macrophage-like cells.

In conclusion, we showed that AR binds the DNA at enhancer elements in macrophage-like cells through the AP-1 complex. The TREM-1 signalling pathway was the dominant biological process under control of AR.

**AR-mediated TREM-1 expression increases chemokine production.** Our previous data suggest a direct AR-mediated regulation of the TREM-1 signalling pathway in macrophages. It was previously established that TREM-1 signalling plays a key role in the production of inflammatory cytokines and chemokines in myeloid cells[23]. In our IPA analysis, 38 genes in the TREM-1 pathway were predicted to be regulated by AR (Supplementary Table 1). To assess whether AR directly modulates the TREM-1

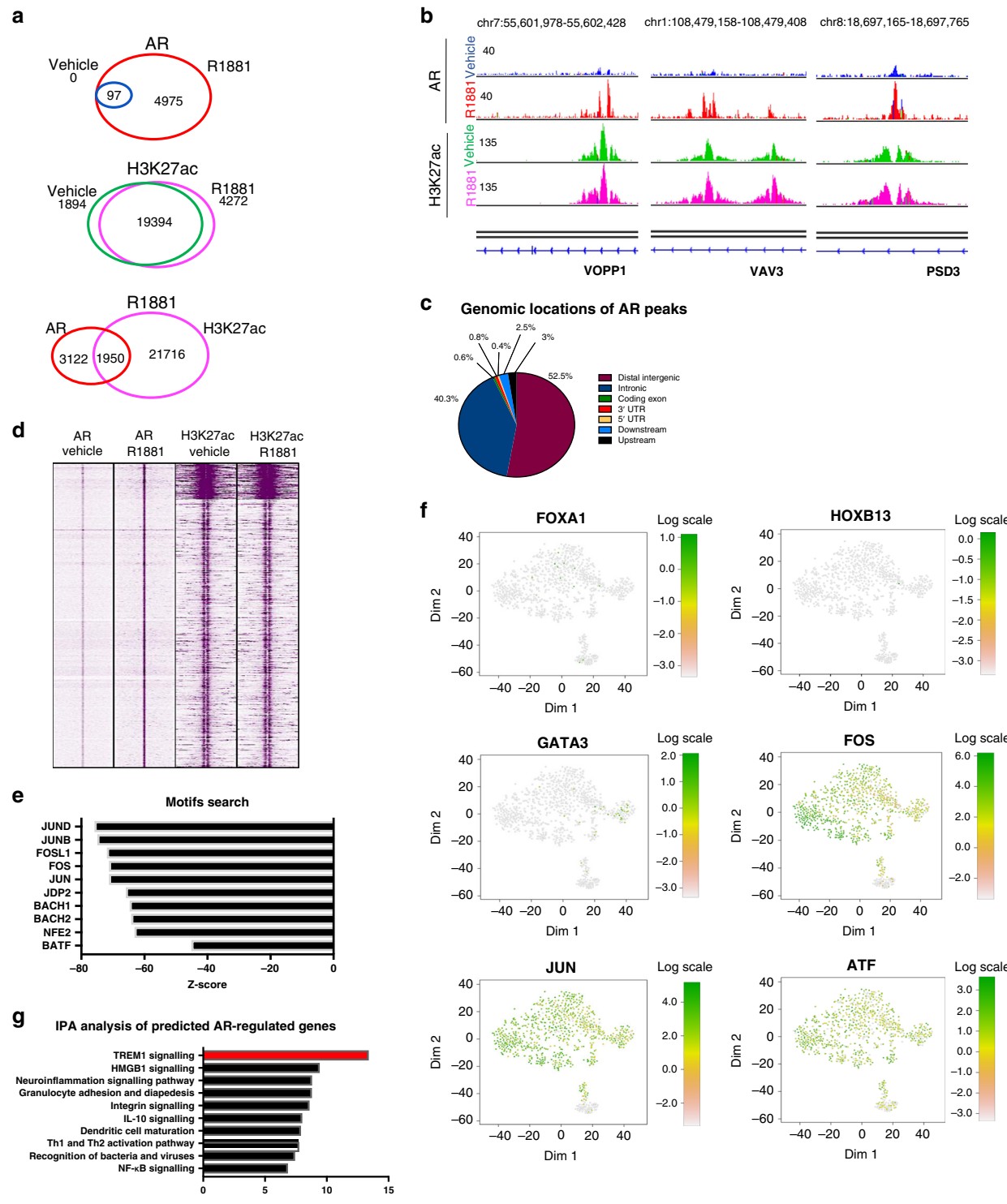

**Fig. 5 ChIP-seq analysis shows distinct AR-binding profiles in THP-1 cells. a** Venn diagrams showing the level of overlap between AR peaks in vehicle and R1881 conditions (upper panel), the overlap between H3K27ac peaks in vehicle and R1881 conditions (middle panel) and the overlap between AR and H3K27ac peaks in R1881 conditions (lower panel) in THP-1[PMA;IFNG;LPS] cells. **b** Snapshot of AR and H3K27ac peaks in THP-1[PMA;IFNG;LPS] cells. Genomic coordinates, gene name and tag counts are indicated. AR peaks in vehicle and R1881 conditions are depicted in blue and red, respectively. H3K27ac peaks in vehicle and R1881 conditions are depicted in green and purple, respectively. Range of normalized read counts is shown on the y axis. **c** Genomic distribution of AR-binding sites relative to the most proximal gene in THP-1[PMA;IFNG;LPS] cells. **d** Clustered heatmap depicts all the AR-binding sites in THP-1[PMA;IFNG;LPS] cells vertically aligned, within a 5 kb window. H3K27ac peaks are shown for the same genomic locations. **e** Motif analysis of AR-binding sites in THP-1[PMA;IFNG;LPS] cells identifies members of the AP-1 complex, including FOS and JUN. Z-score of enrichment is shown on the x axis. **f** RNA expression of AR transcriptional activators in single CD14+ and/or CD11b+ cells isolated from PCa biopsies. In the tSNE plot, every dot represents a single cell. **g** Ingenuity pathway analysis of genes most proximal to AR-binding sites in THP-1[PMA;IFNG;LPS] cells identifies TREM-1 as the most-enriched signalling pathway (indicated in red).

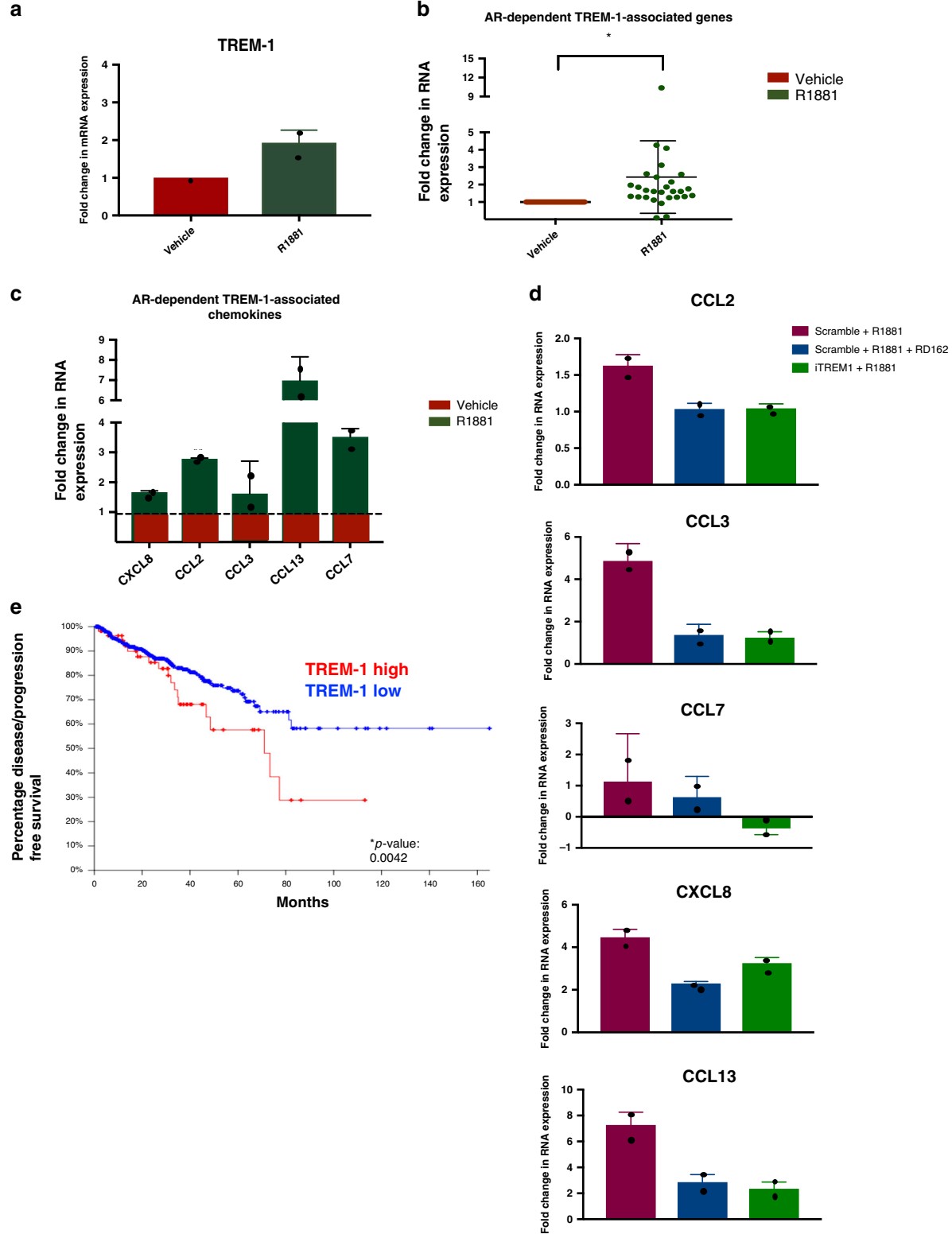

signalling pathway, we evaluated the expression of genes involved in the same pathway. Expression of 28 out of 38 genes was evaluated by qPCR (only previously described TREM-1 associated genes were analyzed). Expression of the *TREM-1* gene (Fig. 6a) and cumulative expression of all the 28 TREM-1 signalling-associated genes (Fig. 6b) in THP-1$^{PMA;IFNG;LPS}$ cells showed a significant increase upon R1881 stimulation compared to vehicle condition. As shown in Supplementary Fig. 8, most of the

TREM-1 associated genes showed increased fold change expression upon R1881 stimulation compared to vehicle control (set to 1), suggesting that the TREM-1 pathway was upregulated in response to R1881. Interestingly, among these genes multiple well-known migration promoting chemokines[24–28] were found to be increased upon R1881 stimulation, including *CCL2, CCL7, CXCL8* and *CCL13* (Fig. 6c). Expression of these chemokines seemed to be under the direct control of TREM-1, as blocking of

**Fig. 6 AR regulates expression of TREM-1 associated genes in THP-1 cells. a** RT-QPCR analysis to assess fold change expression of *TREM-1* (2^-ΔΔCt) in THP-1$^{PMA;IFNG;LPS}$ cells upon R1881 stimulation compared to vehicle control (set at 1) and normalized to *TBP* expression. Datapoints show the mean value of three technical replicates in each experiment, while error bars show the s.e.m. of two independent experiments. Source data are provided as a source datafile. **b** RT-QPCR analysis to assess the fold change expression (2$^{-ΔΔCt}$) upon R1881 exposure of 28 TREM-1-associated genes in THP-1$^{PMA;IFNG;LPS}$ cells, relative to *TBP* expression and normalized to vehicle conditions (set at 1). Each datapoint represents the mean relative gene expression in three technical replicates. The mean fold change expression (vehicle over R1881) of all genes is indicated, while error bar show the s.e.m. of two independent experiments. *: $p = 0.001$. *p*-Value for the comparison of fold change of all genes separately upon vehicle exposure (set at 1) versus R1881 exposure, calculated using an unpaired parametric Student's *t* test with a cutoff for significance of 0.05. Source data are provided as a source datafile. **c** RT-QPCR analysis to assess fold change expression (2$^{-ΔΔCt}$) upon R1881 exposure of 5 TREM-1 associated chemokines in THP-1$^{PMA;IFNG;LPS}$ cells, relative to *TBP* expression. Green bars show R1881 conditions normalized to vehicle (red box and dotted line). Datapoints show mean values and error bars the s.e.m. of two independent experiments with three technical replicates each. Source data are provided as a source datafile. **d** Fold change expression (2$^{-ΔΔCt}$) of TREM-1 associated chemokines in THP-1$^{PMA;IFNG;LPS}$ cells normalized to *TBP* expression. Cells were exposed to scramble peptide or to the inhibitory TREM-1 peptide (iTREM1) in combination with R1881 and/or RD162. Values were normalized to scramble peptide alone. Datapoints show mean values and error bars the s.e.m. of two independent experiments with three technical replicates each. Source data are provided as a source datafile. **e** Kaplan–Meier curves of disease/progression-free survival in PCa patients in relation to levels of *TREM-1* expression (TCGA database, No. of patients = 491). Z-score of ± 1. Fifty-seven cases with a high *TREM-1* expression (red line; 18 relapses/disease progressions) and 434 cases with a low *TREM-1* expression (blue line; 73 relapses/disease progressions). $P = 0.0042$. Log-rank chi square test was used to calculate the *p*-value with a cutoff for significance of 0.05. Source data are provided as a source datafile.

this signalling cascade in R1881 stimulated THP-1$^{PMA;IFNG;LPS}$ cells with the TREM-1 inhibitory peptide LP17 decreased their R1881 induced expression, compared to scramble peptide and R1881 (Fig. 6d). However, AR and H3K27ac binding sites were found within, or proximal to the *TREM-1* gene and multiple *TREM-1* related gene loci, which suggests also direct transcriptional control by AR (Supplementary Fig. 9).

Our in vitro findings were evaluated in the scRNA-seq data of human PCa-associated CD14+ and/or CD11b+ cells. While AR is expressed at the protein level in macrophage-like cells, no *AR* expression was detected in CD14+ and/or CD11b+ cells, which is in line with the absence of AR transcripts in the only other single-cell mRNA sequencing dataset of human tissue-resident macrophages[29].

CD14+ and/or CD11b+ cells express *TREM-1* and its associated chemokines CCL2, CCL3, CCL7, *CXCL8* and *CCL13* (Supplementary Fig. 6E). A correlation, albeit weak, was found between *TREM-1* expression and expression of the prominent chemokine *CCL2* (Pearson correlation coefficient R2: 0.446) (Supplementary Fig. 6F).

Ultimately, we explored the clinical relevance of prostatic *TREM-1* expression using publicly available transcriptomic and clinical data, generated by The Cancer Genome Atlas (TCGA). More specifically, *TREM-1* expression was explored in localized PCa in relation to Disease Free Survival (DFS) in the TCGA database. Out of 491 available untreated primary PCa samples, 57 (11.6%) showed higher *TREM-1* expression (Z-score ±1, difference in standard deviation from the mean). Kaplan–Meier analysis for DFS showed that patients with high *TREM-1* expression in their localized PCa had a significant shorter DFS as compared to patients with low *TREM-1* expression ($p = 0.0042$) (Fig. 6e). Also, with a more stringent z-score of ±2 a significant relation between *TREM-1* expression and DFS ($p = 0.002$) was established (Supplementary Fig. 10). Since *TREM-1* is only expressed in immune cells, and most predominantly in myeloid cells[30], these results suggest a crucial role of these cells in supporting PCa progression.

Taken together, we showed that AR in macrophage-like cells is a regulator of the expression of various chemokines via the TREM-1 signalling pathway, and that increased levels of *TREM-1* expression in localized human PCa correlate with poor outcome.

**TREM-1 signalling in THP-1 cells supports PCa cell migration.** As TREM-1 signalling regulates the expression of chemokines, we evaluated whether TREM-1 in macrophage-like cells was directly

involved in modulating the migration of PCa cells. Migration was estimated in scratch assays, in which CWR-R1 cells were cultured in CM of THP-1$^{PMA;IFNG;LPS}$ cells. As shown in Fig. 7a and quantified in Fig. 7b, migration of CWR-R1 cells was significantly reduced in CM of TREM-1 blocked THP-1$^{PMA;IFNG;LPS}$ cells, as compared to PCa cells cultured in CM of vehicle-treated THP-1$^{PMA;IFNG;LPS}$ cells or scramble control. In addition, in a trans-well assay we show that the capacity of CWR-R1 cells to migrate through a membrane and invade a matrigel layer was also reduced when cultured in CM of TREM-1 blocked THP-1$^{PMA;IFNG;LPS}$ cells (Fig. 7c, d and Supplementary Fig. 11).

We next evaluated whether chemokines associated with the TREM-1 signalling pathway as identified by the ChIP-seq analyses were responsible for the observed increased migration and invasion of PCa cells. As shown in Fig. 7e and quantified in Fig. 7f, blockade of any of the chemokines decreased migration of PCa cells when cultured in THP-1$^{PMA;IFNG;LPS}$ cells CM, while blockade of all chemokines combined had the largest effect. The complete transwell plate used for the migration assay is shown in Supplementary Fig. 12. Subsequently, the influence of the chemokines alone and in combination on PCa cell invasion was evaluated. A decreased PCa invasion was observed when chemokines were blocked individually in THP-1$^{PMA;IFNG;LPS}$ cells CM, while the largest reduction of invasion was found when all evaluated chemokines were inhibited simultaneously (Supplementary Fig. 13A and B).

To evaluate whether PCa cells also express receptors for the identified chemokines, we determined the expression levels of C-C chemokine receptors (CCR) 1-5. Immunohistochemical staining was performed on tissue microarrays (TMAs). No staining was found for CCR1, 2 and 5, while CCR3 and CCR4 staining was found in human PCa cells. Supplementary Fig. 14 shows representative examples of CCR3 and CCR4 receptor staining, which was found to be absent in normal prostate tissue (100 evaluable cores), but expressed—albeit at low levels—in primary PCa (101 evaluable cores) and pelvic lymph node metastases (71 evaluable cores).

To conclude, these results suggest that chemokines under control of TREM-1 support PCa cell migration and invasion. Chemokine receptors are expressed in primary human PCa samples and in lymph node metastases.

**Testosterone affects differentiation of TAMs.** In the above described results, we demonstrated that AR signalling in macrophage-like cells activates TREM-1 signalling which

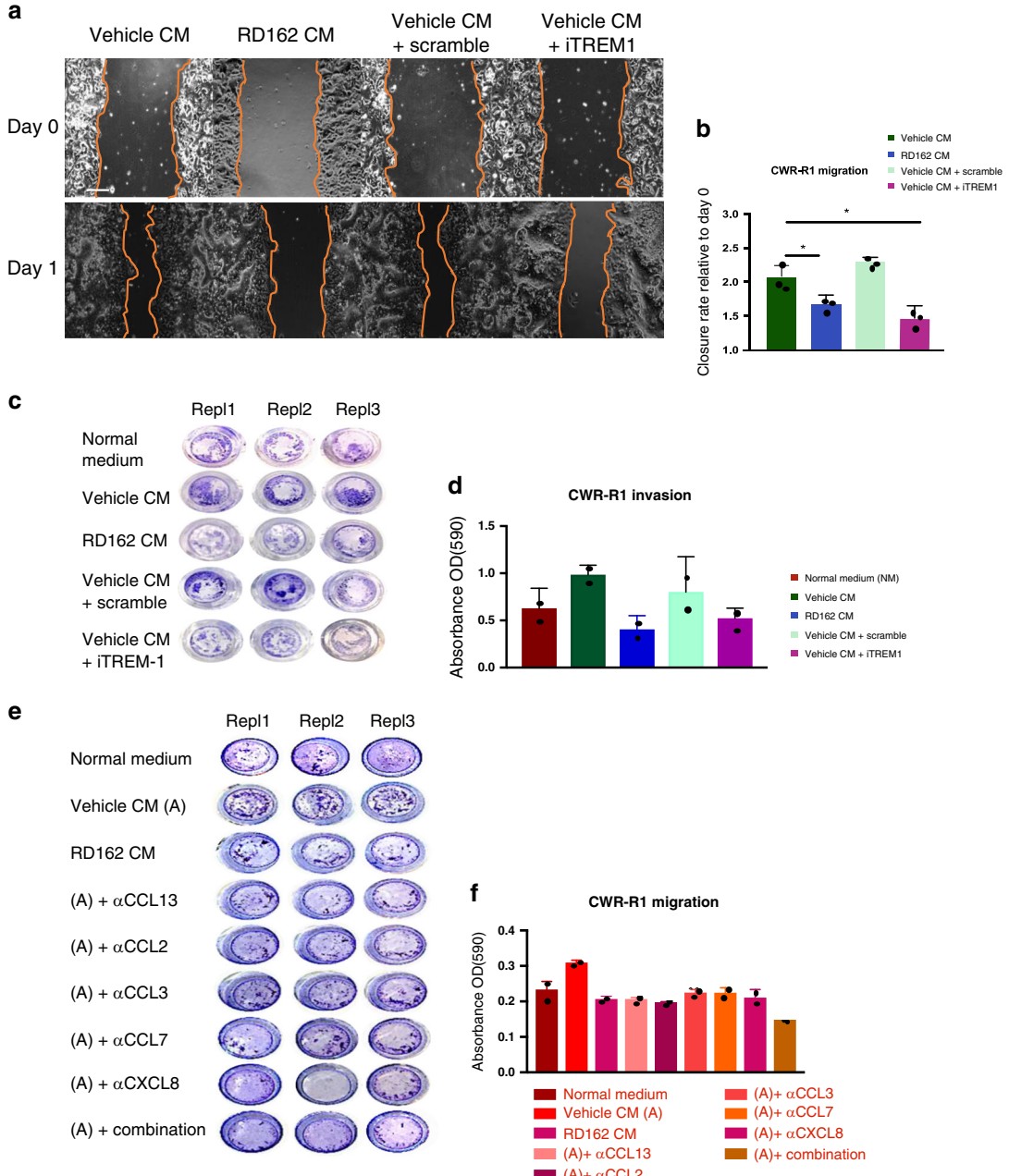

**Fig. 7 TREM-1 signalling in THP-1 cells promotes PCa migration and invasion. a** Representative images of three independent scratch assays of CWR-R1 cells cultured in CM of THP-1[PMA;IFNG;LPS] cells stimulated with vehicle, RD162, scramble peptide or inhibitory TREM-1 peptide (iTREM-1). Three technical replicates were included in each experiment. Scale bar = 200 μm. **b** Quantification of three independent migration scratch assays. Closure of the scratch by CWR-R1 cells after 24 h relative to day 0. Datapoints show mean values of three technical replicates in each experiment. Error bars represent the s.e.m. *: $p = 0.05$, one-way Anova test was used to calculate the $p$-value with a cutoff for significance of 0.05. Source data are provided as a source datafile. **c** Representative images of two transwell invasion assays (three technical replicates) of CWR-R1 cells cultured in normal medium alone or in a combination with CM of THP-1[PMA;IFNG;LPS] cells stimulated with vehicle, RD162, vehicle CM with scramble peptide or vehicle CM with TREM-1 inhibitory peptide. Cells that invade the matrigel and passed through the membrane after 72 h of culture were stained with crystal violet. **d** Quantification of the transwell invasion assays. Invasion was quantified by optical density (OD) of crystal violet stained cells. Datapoints show mean values and error bars the s.e.m. of two independent experiments with three technical replicates each. Source data are provided as a source datafile. **e** Representative images of two transwell migration assays (three technical replicates) of CWR-R1 cells cultured in normal medium alone or in combination with CM of THP-1[PMA;IFNG;LPS] cells stimulated with vehicle or RD162. Additionally, CWR-R1 cells cultured in normal medium in combination with CM of vehicle stimulated THP-1[PMA;IFNG;LPS] cells alone or supplemented with blocking antibodies against CCL13, CCL2, CCL3, CCL7, CXCL8 or the combination of all. Cells that passed through the membrane after 72 h of culture were stained with crystal violet. **f** Quantification of the migration transwell assays. Invasion was quantified by optical density (OD) of crystal violet stained cells. Datapoints show mean values and error bars the s.e.m. of two independent experiments with three technical replicates each. Source data are provided as a source datafile.

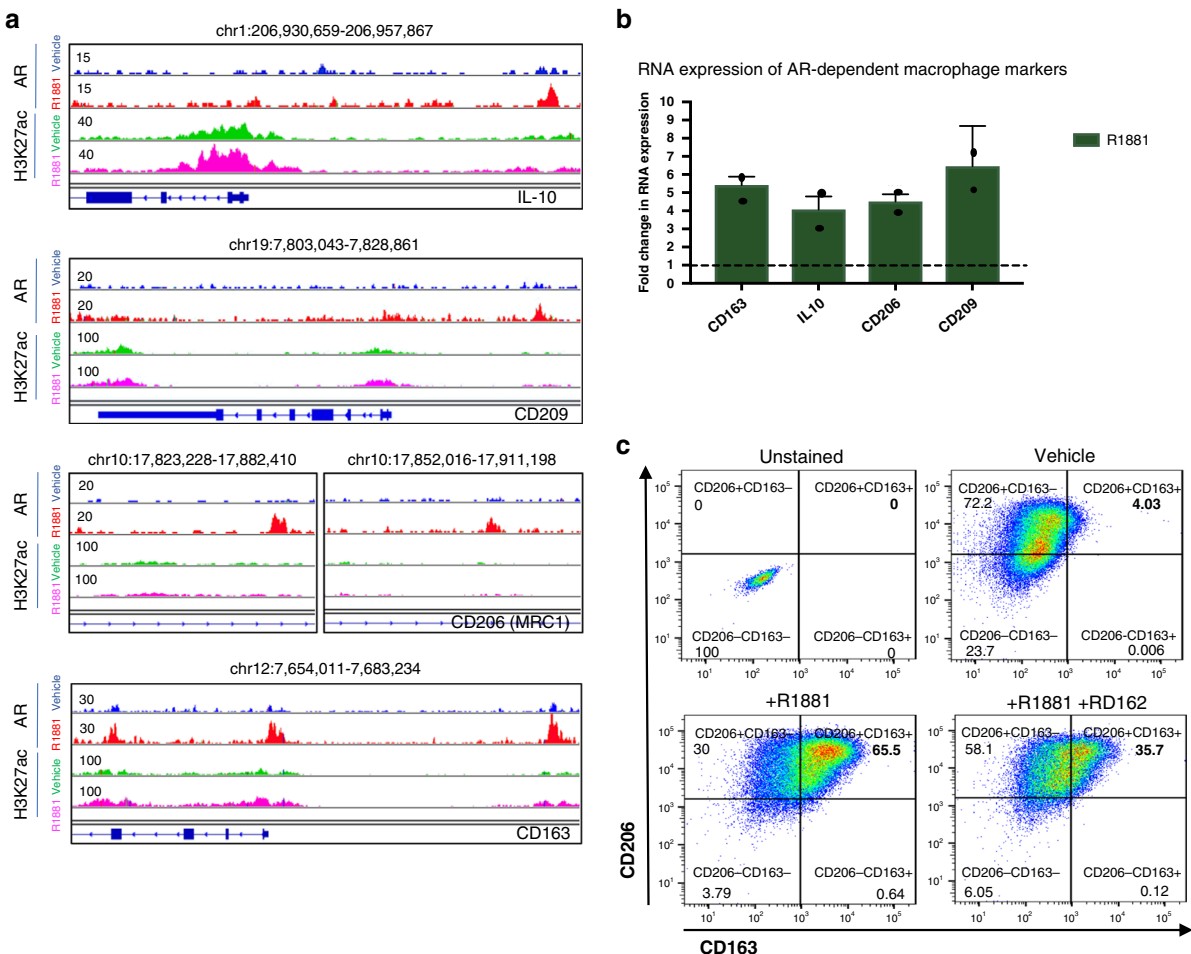

**Fig. 8 AR signalling upregulates tissue macrophage-related markers in THP-1 cells. a** Snapshot of AR and H3K27ac sites proximal to genes associated with macrophage differentiation in THP-1$^{PMA;IFNG;LPS}$ cells (IL-10, CD209, CD206 and CD163). Genomic coordinates, gene name and tag counts are indicated. AR peaks in vehicle and R1881 conditions are depicted in blue and red, respectively. H3K27ac peaks in vehicle and R1881 conditions are depicted in green and purple, respectively. Range of normalized read counts is shown on the y axis. **b** RT-QPCR analysis to assess fold change expression ($2^{-\Delta\Delta Ct}$) of *CD163, IL-10, CD206, CD209* in THP-1$^{PMA;IFNG;LPS}$ cells upon R1881 stimulation, relative to GAPDH expression and normalized to vehicle conditions (dotted line). Datapoints show mean values and error bars the s.e.m. of two independent experiments with three technical replicates each. Source data are provided as a source datafile. **c** Flow-cytometry analysis of surface markers CD206 and CD163 expression in MDM cells. Unstained cells (upper-left) were used as negative control. Twenty-four hrs vehicle stimulated (upper-right), R1881 ($10^{-10}$ M) stimulated (lower-left) and simultaneous R1881 ($10^{-10}$ M) and RD162 ($10^{-7}$ M) stimulated cells (lower-right). The percentage of double positive CD163+/CD206+THP-1$^{PMA;IFNG;LPS}$ cells in the various conditions is represented in bold. Plots are representative for three independent experiments.

stimulates PCa-derived cancer cell migration and invasion. There is abundant literature that supports a role of TAMs in PCa development and progression[8,10,31,32]. Therefore, we evaluated whether AR signalling not only promotes *TREM-1* expression but also TAM differentiation. As shown in Fig. 8a, ChiP-seq analysis showed AR-binding proximal to well-known marker genes of tissue-resident macrophages, including *CD163, MRC1 (CD206), IL-10* and *CD209*. Expression of these genes was increased upon R1881 stimulation (Fig. 8b), suggesting a direct regulation of these genes by AR activation. Although frequently used for TAM identification, these markers are not unique to TAMs, as they are also expressed in normal tissue macrophages. Finally, we used flow-cytometry analysis to evaluate the protein levels of CD163 and CD206 in MDMs (Fig. 8c). The percentage of double positive CD163 and CD206 cells was strongly increased upon R1881 stimulation, while the AR signalling blocker RD162 partially restored the initial expression levels. More details on the gating strategy can be found in Supplementary Fig. 15.

These findings were supported by the scRNA-seq data of human PCa-associated CD14+ and/or CD11b+ cells. As expected, macrophage-like cells taken from the tumorous side of the prostate expressed *CD206, CD163, IL-10* and *CD209*, which are commonly increased in anti-inflammatory M2-like macrophages, while all M1 markers were expressed at very low levels (Supplementary Fig. 5C and D).

In conclusion, our results suggest that AR signalling in macrophage-like cells promotes TAM differentiation.

**Anti-androgen therapy affects TAMs in PCa patients**. We found that AR signalling in THP-1$^{PMA;IFNG;LPS}$ cells regulates markers expressed at high level, but not solely, in TAMs. Therefore, we next evaluated whether anti-androgen treatment of patients affects macrophage differentiation in the PCa microenvironment, which might influence the course of the disease. The effect of anti-androgen therapy on levels of PCa-associated macrophages expressing CD163 was assessed in human prostatectomy specimens. The CD206 marker was not included since CD206 staining was also found in pericyte-like cells and to some extend in endothelial cells, as previously reported[33].

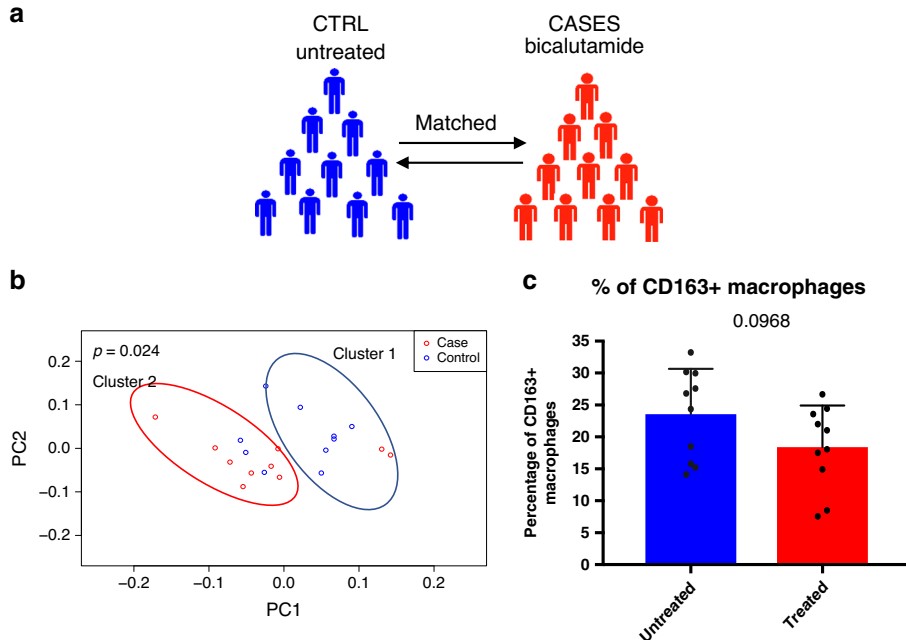

**Fig. 9 Anti-androgen treatment reduces the number of TAMs in the prostate. a** Graphic visualization of the study design: 10 untreated PCa patients and 10 PCa patients treated with the anti-androgen bicalutamide prior to prostatectomy were matched for Gleason pathology score, serum PSA level, age and TNM classification. **b** PCA visualization of the Kmean unsupervised cluster analysis. Group 1 and Group 2 classify patients in untreated and treated patients, respectively. Numerical data in Supplementary Table 3. $P = 0.024$, Pearson correlation test was used to calculate the $p$-value with a cutoff for significance of 0.05. **c** Quantification of the percentage of CD163+ TAMs in the entire population of HLA-DR+ and/or CD14+ cells in prostatectomy specimen of untreated and bicalutamide treated patients. Datapoints show individual patients and error bars show the s.e.m. $p = 0.0968$. Paired parametric Student's $t$ test was used to calculate the $p$-value with a cutoff for significance of 0.05. Source data are provided as a source datafile.

FFPE prostatectomy specimens of 10 PCa patients treated with the anti-androgen bicalutamide prior to surgery and 10 matched untreated PCa patients were collected (Fig. 9a), of whom the age, tumour and treatment characteristics are listed is Supplementary Table 2.

Multiplex immunofluorescence staining was performed to quantify CD163, HLA-DRA and CD14 expression in the tumour area annotated by AMACR staining (Fig. 1b, c). Numbers of PCa-infiltrating CD163+ cells in both cohorts were evaluated by unsupervised K-means cluster analysis as shown in Fig. 9b. Examples of cells scored as CD163 true positive or false positive are shown in Supplementary Fig. 16. All the different combinations of CD163+ cell populations were included in the analyses (e.g.: CD163+HLA-DR+, AR+HLA-DR+, AR+CD14+), which resulted in the identification of two separate patient clusters, that significantly resembled the untreated and treated patient cohorts ($p = 0.024$). More specifically, as shown in Supplementary Table 3, 7 out of 10 patients found in cluster 1 were untreated (group 1), and 8 out of 10 patients found in cluster 2 were treated (group 2). In addition, as shown in Fig. 9c and Supplementary Table 4, bicalutamide-treatment resulted in a non-significant ($p = 0.1$) change in percentage of CD163 expressing HLA-DR+ and/or CD14+ cells compared to their matched controls. As discussed earlier, CD163 is not uniquely expressed in TAMs, but is expressed at a higher level in anti-inflammatory TAMs compared to pro-inflammatory macrophages[34].

Altogether, in this study we unravelled the molecular mechanisms by which AR signalling in macrophage-like cells controls the expression of TREM-1 regulated chemokines, which affect PCa cell migration and invasion. Moreover, anti-androgen therapy might affect TAM differentiation in the PCa microenvironment.

## Discussion

The TME is composed of various cell types, including mesenchymal cells and resident and infiltrating immune cells. It is considered a well-established regulator of the development and progression of many tumour types, including PCa[35–37]. In the presence of PCa, various cells of the TME can alter their phenotype into a malignant state which supports survival of PCa cells[38]. The TME has gained interest over the last decades, as targeting components of the TME could potentially optimize the efficacy of current therapies[39–41].

Total numbers of infiltrating immune cells, including macrophages, have been shown to be prognostic for PCa development[42], while levels of TAM infiltration were predictive for malignancy grade, tumour size and disease recurrence[8] and associated with extra capsular tumour extension[43]. Moreover, myeloid-derived suppressor cells were reported to be crucial for the development of lethal castration resistant disease in a mouse model of PCa, mediated by IL-23[44]. Although this report studied a more advanced stage of PCa as compared to our study, this work underlines the relevance of TME cells of myeloid lineage for PCa progression.

Like epithelial PCa cells, many stromal cells express AR[45,46]. However, expression of AR in specific stromal cells in relation to disease progression is not well documented. The genomic actions of AR in fibroblasts were recently described by us and others groups[12,21,22], but there is very little data on the significance of AR expression in macrophages for cancer progression.

One study in PTEN$^{+/-}$ Macrophage-AR knockout (M-ARKO) mice explored the role of AR signalling in macrophages in prostate tumourigenesis. These mice developed significantly less preneoplastic prostatic intraepithelial neoplasia (PIN) lesions compared to AR-proficient controls, which was mediated through

CCL4-STAT3 signalling[13]. These results are in contrast to the results of a study showing that AR silencing in THP-1 cells supported CCL2-mediated PCa cell migration in vitro[9]. Although these studies suggested a relation between AR signalling in macrophages and PCa development, the genomic and functional mechanisms of AR in macrophages remained unknown.

In this study, we present the genomic and functional role of AR signalling in macrophage-like cells in relation to human PCa cell line migration and invasion. ChIP-sequencing showed that AR in THP-1 cells and MDMs bound the DNA predominantly at enhancer regions via the AP-1 complex, which is in contrast to epithelial PCa cells where AR binding to the DNA is mediated by pioneering transcription factors. IPA revealed that the TREM-1 signalling pathway was the most-enriched biological process in both THP-1 cells and MDMs when exposed to R1881. Previous studies showed that TREM-1 signalling plays a critical role in the production of inflammatory cytokines and chemokines in myeloid cells. TREM-1 is a transmembrane receptor, member of the TREM immunoglobulin family and was discovered in 2000 by Bouchon et al.[30]. Expression was found to be limited to innate immune cells and predominantly to monocytes and macrophages, and it was described to be critically involved in the modulation of the immune response via interaction with its adaptor protein DNAX-activating protein 12 (DAP12). Phosphorylation of DAP12 resulted in the activation of downstream signalling molecules including the MAP/ERK pathway. Importantly, these molecules also regulate the activation of NF-kB, a master regulator of the expression of key inflammatory genes[47]. Amplification of the inflammatory response is the best-characterized function of the TREM-1 receptor. Cytokines including TNF-α, CCL2 (MCP-1) and IL-1 were described to be upregulated upon over-activation of the TREM-1 receptor in monocytes[48]. Following this observation, other studies further explored the relevance of TREM-1 signalling in modulating the immune response[49–52].

We described that expression of the TREM-1 gene and most of its downstream associated cytokines and chemokines were increased upon R1881 stimulation in THP-1 cells, which was associated with increased migration and invasion of human PCa cells. However, blockade of TREM-1 in THP-1 cells did not result in decreased expression of all TREM-1 associated cytokines. Although we did find AR binding at the loci of various genes, other transcription factors might also control the expression of TREM-1 associated cytokines. Moreover, the protein LP17 was used as an inhibitor of TREM-1 signalling. Consequently, we cannot rule out that other signalling cascades may be affected by LP17 as well.

Our results suggest that activation of AR in macrophages increases the metastatic potential of PCa cells. Question is, how do these in vitro results translate to primary and metastatic PCa? The gradient of chemokines secreted by the resident macrophages in the PCa microenvironment might as well maintain the cancer cells in the local tumour. Chemokine receptors are essential for the chemotactic recruitment of PCa cells as observed in the in vitro migration and invasion assays. Expression of CCR3 and CCR4 was found in primary PCa cells and metastatic PCa cells, while expression was absent in normal prostate cells. In contrast, expression of CCR1, CCR2 and CCR5 was absent. However, chemokine receptors have little chemokine specificity[53].

Expression of TREM-1 was described to promote tumourigenesis and support tumour growth in various tumour models including intestinal-[54], pancreatic- and lung cancer[55,56]. In non-small cell lung cancer (NSCLC) patients, TREM-1 expression in TAMs was associated with tumour recurrence and poor survival[57]. Co-culture of blood monocytes of these NSCLC patients with lung cancer cells resulted in TREM-1 upregulation in the monocytes, and in the same study inhibition of TREM-1 expression by shRNA was associated with decreased invasiveness of lung cancer cells. In another study in NSCLC, TREM-1 expression was shown to be induced in TAMs by tumour cell mediated prostaglandine-2 (PGE2) production, suggesting a link between the tumour cell derived cyclo-oxygenase 2 (COX-2) pathway and the expression of TREM-1 in TAMs[58]. Consistent with the present report, these studies suggest a cross-talk between cancer and TREM-1 signalling in myeloid cells.

We have shown that AR signalling in THP-1 cells promotes the expression of TAMs associated genes. Moreover, in our study into PCa samples, a trend towards decreased CD163+ macrophage infiltration was observed in treated patients compared to untreated patients. Studies in mice also showed that androgens enhance TAM polarization in vivo and in vitro[59,60]. Moreover, alveolar macrophages lacking AR were shown to express less TAM markers as well as a lower chemokine production then AR-proficient macrophages, supporting our findings[61].

In conclusion, in this study we provide evidence on how AR signalling in macrophage-like cells affects macrophage differentiation and induces the expression of TREM-1 mediated chemokines, resulting in enhanced migration of PCa-derived cells. These results suggest that AR signalling in macrophages might affect human PCa progression. Furthermore, we showed how off-target effects of antihormonal therapy can synergize to reduce PCa progression.

## Methods

**Ethics statement and clinical samples.** Buffy coats of peripheral blood samples from healthy donors were collected from the Sanquin Blood Supply Foundation in Amsterdam, permitted by the Minister of Health, welfare and Sport (VWS). Post-surgical tumour biopsies of robotic-assisted laparoscopic prostatectomies (RALP) of three untreated PCa patients (Gleason score 3 + 4) were freshly collected for single-cell flow cytometry (FACS) sorting. FFPE prostatectomy specimens of 10 untreated and 10 neoadjuvant bicalutamide treated patients, who underwent a RALP between 2003 and 2013 were retrieved from the NKI bioarchive. Patients were treated with bicalutamide for 12–20 weeks in doses between 50 and 150 mg daily. Untreated and treated patients were individually matched for Gleason score, initial PSA, age and pT classification. Inclusion criteria were: Gleason score 6-7-8, pT classification 2-3, PSA 0–50 ng ml$^{-1}$ and age between 51 and 70 years (Supplementary Table 2). The use of archival prostatectomy material and biopsies from fresh prostatectomy specimens for research purposes at the Netherlands Cancer Institute have been executed pursuant to Dutch legislation and international standards. Prior to 25 May 2018, national legislation on data protection applied, as well as the International Guideline on Good Clinical Practice. From 25 May 2018 on, we also adhere to the General Data Protection Regulation. Within this framework, patients are informed and have always had the opportunity to object or actively consent to the (continued) use of their personal data & biospecimens in research. For the current studies, informed consent was obtained from all patients. Hence, the procedures comply both with (inter-)national legislative and ethical standards.

**Isolation of CD14+ and/or CD11b+ cells from prostate biopsies.** Freshly collected biopsies were chopped in phosphate buffered saline (PBS) and processed in a gentleMACS Dissociator (Miltenyi Biotec). Program used for human tissue were 'h_tumour_01' for one time and 'h_tumour_03' for three times. The cell suspension was subsequently filtered using a 70 μm nylon cell strainer (BD Biosciences) and washed with PBS. Cells were centrifuged at 1200 rpm for 6 min and resuspended in PBS. Cell suspension was stained for flow cytometry sorting with CD45-APC, CD3-FITC, CD14-PE and CD11b-PE antibodies (eBioscience) in PBS + 0.5% bovine serum albumin (BSA) for 20 min at 4 °C and sorted in 384 well plates with a MoFlo Astrios Beckman Coulter.

**Single-cell sequencing of CD14+ and/or CD11b+ cells.** Sorted cells were lysed at 65 °C for 5 min, followed by cDNA synthesis. Second strands were dispersed with the Nanodrop II liquid handling platform (GC biotech). The aqueous phase was collected, followed by IVT transcription for library preparation using the CEL-Seq2 protocol[62]. 384 cell barcodes containing a 6 bp UMI and mineral oil (Sigma Aldrich) was used. Liquid handling was performed by the Nanodrop II (GC Biotech) and Mosquito®HTS (TTP labtech) platforms. Cells with less than 2500 unique transcripts were discarded, and only genes that were detected with more than three unique transcripts in at least two cells were selected. All analyses were performed using the RaceID2 algorithm[63].

**Immunohistochemistry**. Immunohistochemistry of FFPE tumour samples was performed on a BenchMark Ultra autostainer (Ventana Medical Systems). Briefly, paraffin sections were cut at 3 μm, heated at 75 °C for 28 min and deparaffinised in the instrument with EZ prep solution (Ventana Medical Systems). Heat-induced antigen retrieval was carried out using Cell Conditioning 1 (CC1, Ventana Medical Systems) for 64 min at 95 °C. CCR3 clone Y31 (AbCam) was detected using 1/800 dilution, 1 h at room temperature (RT), and CCR4 polyclonal (Sigma Aldrich) using 1/200 dilution, 1 h RT. Bound antibody was detected using the OptiView DAB Detection Kit (Ventana Medical Systems). Slides were counterstained with Hematoxylin and Bluing Reagent (Ventana Medical Systems).

**Non-automated immunofluorescence staining**. For non-automated immuno-fluorescence staining, MDMs and cells cultured on coverslips were fixed with 4% paraformaldehyde in PBS for 10 min at room temperature (RT) and washed with PBS. Cells were then permeabilized with 0.1% Triton X-100 in PBS and 0.5% BSA for 5 min and blocked with 1% BSA in PBS for 1 h, prior to staining. Anti-AR Ab (Santa Cruz, sc-816, 1:50) and anti-CD68 Ab (Dako, KP1, 1:100) were used as primary antibodies and fluorescent Alexa Fluor 488 anti-mouse and Alexa Fluor 568 anti-rabbit (ThermoFisher Scientific) as secondary antibodies. Fluorescence was assessed using a SP5 Leica Confocal Microscope.

**Automated multiplex staining on Discovery Ultra Stainer**. For automated multiplex immunofluorescence staining, 3 μm slides were cut on DAKO Flex IHC slides. Slides were then dried overnight and stored at 4 °C and used for staining. Prior to multiplex staining 3 μm slides were cut on DAKO Flex IHC slides. Slides were then dried overnight and stored at 4 °C. Before a run was started, slides were baked for 30 min at 70 °C in an oven. Staining was performed on a Ventana Discovery Ultra automated stainer, using the Opal 7-Color Manual IHC Kit (50 slides kit, Perkin Elmer, cat NEL81101KT). Protocol starts with baking for 28 min at 75 °C, followed by dewaxing with Discovery Wash using the standard setting of three cycles of 8 min at 69 °C. Pre-treatment was performed with Discovery CC1 buffer for 32 min at 95 °C, after which Discovery Inhibitor was applied for 8 min to block endogenous peroxidase activity. Specific markers were detected consecutively on the same slide with the following antibodies, Anti-AMACR (Clone 13H4, Cat M3616, Dako, 1/1600 dilution 32 min at RT), anti-AR (Clone SP107, Cat M4074, Spring Bioscience, 1/500 dilution 1 h at RT), anti-CD14 (Clone EPR3653, Cat 114R-14, Cell Marque, 1/50 dilution, 1 h at RT), anti-CD163 (clone 10D6, Cat NCL-CD163, Leica, 1/500 dilution, 1 h at RT), Anti-CD20 (Clone L26, cat M0755, Dako, 1/500 dilution, 1 h at RT), Anti-HLA-DR (Clone TAL.1b5, Cat M0746, Dako, 1/200 dilution, 1 h at RT). Each staining cycle was composed of four steps: Primary Antibody incubation, Opal polymer HRP Ms+Rb secondary antibody incubated for 32 min at RT, OPAL dye incubation (OPAL520, OPAL540, OPAL570, OPAL620, OPAL650, OPAL690, 1/50 or 1/75 dilution as appropriate for 32 min at RT) and an antibody denaturation step using CC2 buffer for 20 min at 95 °C. Cycles were repeated for each new antibody to be stained. At the end of the protocol slides were incubated with DAPI (1/25 dilution in Reaction Buffer) for 12 min. After the run was finished slides were washed with demi water and mounted with Fluoromount-G (SouthernBiotech, cat 0100-01) mounting medium. After staining slides were imaged using the Vectra 3.0 automated imaging system (PerkinElmer). First whole slide scans where made at ×4 magnification. After selection of the region of interest, multispectral images were taken at ×20 magnification. Library slides were created by staining a representative sample with each of the specific dyes. Using the InForm software version 2.3 and the library slides the multispectral images were unmixed into eight channels: DAPI, OPAL520, OPAL540, OPAL570, OPAL620, OPAL650, OPAL690 and Auto Fluorescence and exported to a multilayered TIFF file. The multilayered TIFF's were fused with HALO software version 2.1 to create one file for each sample. Image analysis was then performed using the HALO software module HighPlex FL version 2.0. Tissue annotation and cell identification was performed by a dedicated uro-oncology pathologist. Based on AMACR staining, the Tumour area was identified. Using HALO software a 200 μm margin around the Tumour area was selected as tumour border. All tissue outside this margin was considered Distant.

**Multiplex immunofluorescence analysis**. Python 3.6.3 was used to process the HALO output files. Files contained: patient-number identifier, x/y coordinates, tissue annotation, marker positivity for membrane, cytoplasm or nucleus, and overall positivity. The output of the Python script was processed further in R Version 3.4.3. The final file contained the fractional counts for each phenotype for each patient-number identifier. This resulting data matrix was then transformed with a variance stabilizing function. K-means clustering and the quantification of the percentage of CD163+ of the entire population of HLA-DR+ or/and CD14+ cells was performed using the following subset of phenotypes from the normalized and stabilized dataset: CD163+AR+, CD163+AR+HLA-DR+, CD163+CD14+ HLA-DR+, CD163+CD14+AR+, CD163+CD14+, CD163+CD14+AR+HLA-DR+, CD163+HLA-DR+. Each staining was evaluated together with an experienced pathologist. A threshold was set based on the intensity of staining within the cytoplasm/membrane or nuclear compartments, depending on the specific staining. This threshold was tested on multiple samples and then re-checked by the pathologist. If not correct the threshold was adjusted and rerun until it was correct.

**Generation of monocyte-derived macrophages**. Buffy coat was mixed 1:2 with PBS. The mixture was then added 3:1 to Ficoll gradient (Invitrogen, 17-1440-03) and spun down at 2100 rpm for 25 min at RT (w/o brakes). The leucocyte ring was collected and washed with cold PBS. Cells were then resuspended in MACS buffer (0.5% BSA in PBS) containing anti-human CD14 microbeads (Miltenybiotec). CD14+ cells were cultured in RPMI medium, 10% FBS, 20 ng ml$^{-1}$ GM-CSF (R&D Systems, 215-GM-010) and 1% Pen Strep (Gibco) (FBS-RPMI) for 24 h. Then, medium was replaced with RPMI, 5% DCC, 20 ng ml$^{-1}$ GM-CSF and 1% Pen Strep for 3 days. Cells were then stimulated for 24 h with 10 ng ml$^{-1}$ of IFN-γ (R&D Systems, 285-IF-100) and 10 ng ml$^{-1}$ of LPS (ENZO, Life Science O55:B5) for differentiation into macrophage-like cells.

**Cell lines and hormones**. THP-1 (human monocytic cell line from acute mono-cytic leukaemia), M14 (melanoma cell line), CWR-R1, PC3 and LNCaP (all PCa cell lines) were cultured in RPMI, 10% FBS and 1% Pen-Strep. All cell lines were purchased from ATCC and routinely tested for mycoplasma infection. THP-1 cells were stimulated with 100 ng ml$^{-1}$ of PMA (Sigma, P1585) for 2 days followed by 10 ng ml$^{-1}$ IFN-γ and 10 ng ml$^{-1}$ of LPS for 24 h. Cells were cultured in DCC-RPMI medium for 3 days prior to stimulation with vehicle (DMSO), testosterone analogue R1881 (Sigma, R0908) (range $10^{-8}$–$10^{-10}$ M) alone or in combination with anti-androgen RD162 (Axon 1532) (range $10^{-6}$–$10^{-8}$ M).

**TREM-1 inhibitors and blocking antibodies**. LP17 TREM-1 inhibitory peptide (LQVTDSGLYRCVIYHPP) and LP17 scramble protein (TDSRCVIGLYHPPL QVY) were chemically synthesized (Pepscan). The TREM-1 inhibitors or scramble peptides were used at a concentration of 200 ng ml$^{-1}$ in FBS-RPMI medium in combination with vehicle or RD162 for 24 h. Cells were then washed with PBS and further cultured in fresh DCC-RPMI medium for 48 h. Medium was collected and used as CM for further experiments.

Blocking antibodies against CXCL8, CCL2, CCL3, CCL13 and CCL7 (all R&D Systems) were used in THP-1 CM at a concentration of 100 ng ml$^{-1}$ for anti-CCL3 and anti-CXCL8, 1 μg ml$^{-1}$ for anti-CCL2, anti-CCL7 and 10 μg ml$^{-1}$ for anti-CCL13.

**Flow cytometry**. Monocyte-derived macrophages and THP-1 cells were stimulated with vehicle, R1881 ($10^{-10}$ M) alone or in combination with RD162 ($10^{-7}$ M) for 24 h. Cells were then collected in FACS buffer (0.5% BSA in PBS) and incubated for 20 min with anti-CD163 APC and anti-CD206 PE (eBioscience) at 4 °C in the dark. Analysis was performed on an LSR Fortessa SORP1 flow cytometer.

**Subcellular fractioning and western blot**. After 4 h stimulation with vehicle or R1881 ($10^{-8}$ M), THP-1 and CWR-R1 cells were harvested for subcellular frac-tioning. Briefly, cells were scraped with PBS and 1X protease inhibitor cocktail (Roche) and centrifuged at 2000g for 7 min. Pellet was resuspended in 200 μL of subcellular fraction buffer (10 mM HEPES, 10 mM KCL, 1.5 mM MgCl$_2$, 0.34 M Sucrose, 10% Glycerol, 1 mM DTT, 0.1% Triton X-100) and centrifuged at 1300g for 5 min. Supernatant (cytoplasmic fraction) was stored at −20 °C. 200 μL of buffer B (3 mM EDTA, 0.2 mM EGTA, 1 mM DTT) was added to the pellet and centrifuged at 1700g for 5 min. Supernatant (nucleoplasmatic fraction) was stored at −20 °C. 200 μL of Laemmli buffer was added to the pellet (chromatin fraction) and stored at −20 °C. Antibodies used for assessments of subcellular AR expression were anti-AR (Santa Cruz, sc-816, 1:1000), with anti-Pol-II (Santa Cruz sc-56767) as loading control.

Whole cell lysates were also collected in RIPA buffer and 1X protease inhibitor cocktail for AR western blots, using anti-AR (Santa Cruz, sc-816, 1:1000), with anti-ß-Actin (Novus Biological, NB600-501, 1:5000) as loading control. The uncropped blots can be found in the Source Data file and in Supplementary Fig. 17.

**CM collection**. For CM collection, THP-1$^{PMA;IFNG;LPS}$ cells and MDMs were stimulated for 8 h with vehicle (DMSO), R1881 (10 nM) or RD162 (10uM). Next, medium was decanted, cells were carefully washed with PBS and fresh DCC-RPMI (hormone-deprived) medium was added. After for 48 h CM was collected and used for further experiments.

**Proliferation, migration and invasion assay**. For proliferation assays, CWR-R1 cells were seeded in 384 well plates (500 cells/well). Thereafter, cells were cultured in hormone-deprived CM from stimulated THP-1$^{PMA;IFNG;LPS}$ cells (Vehicle CM or RD162 CM) mixed (1:4) with fresh 10% FBS-RPMI medium (NM). Proliferation was estimated using an IncuCyte Zoom microscope. For migration assays CWR-R1 or PC3 cells were seeded in 12 well plates. Once confluency was reached, a scratch on the cell surface was made with a 200 μl pipet tip. Next, cells were cultured in hormone-deprived CM from stimulated THP-1$^{PMA;IFNG;LPS}$ cells (Vehicle CM or RD162 CM) mixed (1:4) with fresh 10% FBS-RPMI NM. Hormone-deprived CM (DCC medium) was used as a negative control. Culture was continued until one of the conditions reached a near closure of the scratch. This could vary depending on the batch of CM used. At the starting point (0 h) and endpoint of the scratch assay, migration of the cells was quantified using ImageJ software (1.50i) as number of

pixels in defined areas. Closure rate was assessed comparing final timepoint over the initial timepoint in each condition.

Alternatively, 96 transwell plates with 8.0 μm pores (Corning, CLS3374) or 24 transwell plates with 8.0 μm pores (Corning, CLS3428) were used to assess migration and invasion ability of PCa cells. CM used in these assays consisted of hormone-deprived CM from stimulated THP-1[PMA;IFNG;LPS] cells (Vehicle CM or RD162 CM) mixed (1:4) with fresh 10% FBS-RPMI medium. Moreover, to evaluate the relevance of specific chemokines for migration and invasion of PCa cells, cells were cultured in hormone-deprived CM from THP-1[PMA;IFNG;LPS] cells mixed (1:4) with fresh 10% FBS-RPMI medium containing neutralizing antibodies against CCL2, CXCL8, CCL3, CCL13 and CCL7. For migration assays, CWR-R1 cells were seeded in the upper chamber of the transwell, while CM was added to the bottom. After 48 to 72 h, PCa cells that migrated through the other side of the membrane were quantified using crystal violet. For invasion assays, matrigel (Sigma, E1270) was added to the upper chamber of the transwell before CWR-R1 cells were seeded. Cells that invaded through the membrane were quantified as previously described.

**Chick CAM tumour grafts.** Fertilized chicken White Leghorn eggs were incubated in a fan-assisted hatching incubator at a temperature of 38 °C and constant air humidity of 70%. On Embryonic Developmental Day 6 (EDD6) the CAM surface was gently scratched, and 50 μl of $2 \times 10^6$ PC3 PCa cells suspended in 50% growth factor reduced Matrigel (Becton Dickinson, Breda, The Netherlands) were grafted on the CAM. The eggs were incubated under standard conditions. On EDD10, eggs were treated with 50 μl of either 0.9% NaCl, CM from DMSO-treated THP-1[PMA;IFNG;LPS] cells (DMSO CM), CM from RD162-treated THP-1[PMA;IFNG;LPS] cells (RD162 CM) or CCL2 in PBS supplemented with 0,1% BSA (10 ng ml$^{-1}$). The daily based treatment lasted until EDD14. Tumour volume was followed every day until EDD17, and calculated using an external calliper, by the modified ellipsoid formula ½ × (length × width$^2$). On EDD17, distant (>1.5 cm distance from the tumour) normal CAM was collected and used for RNA isolation and downstream analyses.

**Chromatin immunoprecipitation (ChIP).** Protein–DNA complexes were pre-fixed in solution A (50 mM Hepes-KOH, 100 nM NaCl, 1 nM EDTA, 0.5 EGTA) with 2 mM disuccinimidyl glitarate (DSG) (CovaChem) for 35 min at room temperature, followed by fixation with 1% formaldehyde for 10 min and subsequently quenched with glycine. After three PBS washes, cells were collected in lysis buffer containing proteinase inhibitors (10 nM Tris-HCl, 100 mM NaCL, 1 mM EDTA, 0.5 mM EGTA, 0.1% Na-deoxycholate, 0.5% N-lauroylsarcosine) for nuclei extraction, followed by sonication for at least 10 cycles of 30 s on and 30 s off using a Diagenode Bioruptor Pico. Size of the DNA segments was evaluated on agarose gel. Lysate was then incubated with the specific antibodies overnight. Antibodies used were: Anti-AR (Millipore, 06-680) (7 μg per sample) and anti-H3K27ac (Active Motif, 39133) (5 μg per sample). After incubation, reverse crosslinking was performed at 65 °C overnight. DNA was then treated with 1 mg ml$^{-1}$ RNAseA for 1 h at 37 °C followed by treatment with proteinase K for 2 h at 55 °C. DNA was then collected with phenol-chloroform protocol and submitted for sequencing.

**DNA sequencing, enrichment and data analysis.** DNA was amplified as previously described[64], and processed for library preparation (Part# 0801-0303, KAPA Biosystems kit). An Illumina HiSeq 2500 Genome Analyzer (65-bp reads) was used for sequencing. Alignment of the sequences was performed on Human Reference Genome (assembly hg19, February 2009) and reads were filtered based on MAPQ quality (>20). Peaks called by both DFilter[65] (bs = 100, ks = 50, nonzero) and MACS peak caller ($P = 10^{-7}$)[66] were used for the analysis. For peaks and motif analysis the Cistrome platform was used (cistrome.org). Also, the cis-regulatory element annotation system (CEAS) was used for analysis of the genomic distributions of binding sites. Integrative Genomic View (IGV) and SeqMINER were used for peaks visualization and IPA (QUIAGEN 2015) was used for analysis of predicted AR-target genes. Binding sites found in the gene body or 20 kb upstream from the transcription start site were considered proximal to the gene.

**Quantitative PCR analysis.** Monocyte-derived macrophages and cell lines were stimulated with vehicle (DMSO), R1881 ($10^{-9}$) alone or in combination with RD162 ($10^{-7}$) for 24 h. RNA was isolated in Trizol according to the manufacturer's instructions (Invitrogen, 15596026). For cDNA production, the Tetro cDNA Synthesis Kit was used (Bioline, BIO-65043). SensiFAST$^{TM}$ Real-Time PCR Kits were used for qPCR. Primers used for PCR and qPCR are shown in Supplementary Table 5 (all Invitrogen). For transcript analysis of distant CAM tissues, cDNA was synthesized from 1 μg RNA using iScript (Bio-Rad), according to the manufacturers' instructions. qPCR was performed using SYBR Green Supermix (Bio-Rad). Primers distinguishing between human and chicken transcripts were designed as previously described[67] and shown in Supplementary Table 5. Vimentin expression was evaluated, and normalized to the reference gene cyclophilin A using the 2^-dCt method[67].

**TCGA database analysis.** The website cBioPortal for Cancer Genomic website version 1.12.1 was used to explore the clinical relevance of TREM-1 expression in the primary PCa TCGA database. Patients were divided based on TREM-1 mRNA upregulation (z-score of ±2.0 or ±1). Disease and Progression-free Kaplan–Meier curves were automatically generated.

**Reporting summary.** Further information on research design is available in the Nature Research Reporting Summary linked to this article.

## Data availability

The ChIP-seq data generated in this study have been deposited in the National Center for Biotechnology Information (NCBI) in the Gene Expression Omnibus (GEO) database under accession number GSE131381 and the single-cell RNA-sequencing data under accession number GSE133094. Source data are provided with this paper.

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

## Acknowledgements

The authors thank the Genomics core facility and Core Facility Molecular Pathology and Biobanking of NKI and the Sorting facility of the Hubrecht Institute for their technical support. The authors kindly thank Dr. Yongsoo Kim for bioinformatics support. This work was supported by grants from Marie Curie ITN-TIMCC and KWF Dutch Cancer Society.

## Author contributions

B.C., A.Z., A.M.B. and W.Z. conceived of the presented idea and wrote the manuscript. B.C., A.Z., A.M.B., W.Z., M.H.M.M., J.v.B. and A.W.G. planned the experiments. B.C., J.v.B. and A.Z. carried out the experiments. J.R.v.B. performed the Chick-CAM assay. J.d.J. and J.S. provided pathology review. I.H. and D.P. performed Vectra-based multiplex staining. E.H. and Y.L. performed data analysis and interpretation. M.J.M., J.V. and provided computational support. H.G.v.d.P. provided specimens material. J.P.d.B. provided scientific inputs for the development of the project. All the co-authors critically reviewed the present manuscript before submission.

## Competing interests

The authors declare no competing interests.
