## [Peer Review File · Nature Communications]

Reviewers' comments:

Reviewer #1, expert in AR signaling (Remarks to the Author):

In the manuscript entitled "Androgen Receptor Signalling in Macrophages Promotes TAM Polarization and TREM-1 mediated Prostate Cancer Cell Migration and Invasion", Cioni and co-workers studied androgen receptor (AR) signalling in macrophages, especially whether AR activation regulates prostate cancer migration and invasion. To address these effects, the authors utilized THP-1 monocytic leukemia cells, CWR-R1 and LNCaP prostate cancer cell lines as models for the studies in vitro and fractions of tumour-associated macrophages in the prostate cancer-associated stroma biopsied from patients treated with anti-androgen therapy.

The authors conclude that AR signalling in macrophages supports prostate cancer cell migration and invasion, and that inhibition of AR signalling in macrophages is a newly identified mechanism of action of anti-androgen therapy.

To be able to warrant the statements in the manuscript, major modifications and strengthening of the data are needed. The manuscript is adequately written, but modifications also to the figures are needed to clearly describe the results. The manuscript could be improved by considering the following issues:

Major concerns:

1) Authors conclude that the migration of CWR-R1 cells cultured in THP-1 conditioned medium affects CWR-R1 cell migration due to activated AR signalling in THP-1 cells. However, the authors state that "the genomic actions of AR in fibroblasts were recently described by us and others groups" (page 14, line 334). The CWR-R1 cell line is a co-culture of epithelial cells and stromal fibroblasts. How do the authors excluded that the effects seen on the migration in the THP-1-conditioned medium exposed CWR-R1 cells is not due to activated AR signalling in the fibroblasts of the CWR-R1 cell line?

2) In the Figure 3B, the authors show the results for CWR-R1 cell line cultured with THP-1 conditioned medium. In the materials and methods section, they mention that "conditioned medium was then collected and used 1:4 with FBS-RPMI medium (normal medium) for proliferation, migration and invasion assays" (page 19, line 464). However, the difference between "Normal medium" and "Vehicle CM" is unclear. Is the effect on the cell migration between Normal medium and Vehicle CM seen in Figure 3B and C seen due to the fact that "Normal medium" contains 2.5% more FBS than the "Vehicle CM"? Are there migration differences seen between the "Normal medium" and the "Vehicle CM" conditions without pre-incubation with THP-1 cells? Does the DCC-RPMI alone affect the CWR-R1 cell migration? The medium conditions should be clearly defined in the manuscript.

3) In the figure 6 panels A and B, the Vehicle CM effect on migration of CWR-R1 cells is shown after 3 days. However, the wound closure rate relative to day 0 is totally different from the result seen in the Figure 3 where the wound was almost closed already after one day. In the Figure 6, the closure rate relative to day 1 is 2.0 with vehicle CM at the time of 3 days, whereas in the figure 3B the closure rate under the same conditions is almost 6.0. How was the closure rate measured? How many times were these experiments (Figure 3 and 6) repeated?

4) The amount of biological and technical replicates for the assays nor the star abbreviations for calculations of significance are not clarified throughout the manuscript.

5) Authors stimulated THP-1 cells with PMA and then used LPS and IFN- γ for TAM polarization. LPS and IFN- γ stimulation of M Φ s polarizes M Φ s to M1 type macrophages which are generally thought to prevent the cancer cell locomotion and tumor development. In this study, the authors discuss the role of AR in M Φ s in relation to human prostate cancer progression. How are the results of this manuscript related to the findings in the field? Would M Φ polarization to type M2 macrophages have different

effect?

Minor issues:

1) It is very difficult to compare the RNA expression fold changes in the Figure 5. To clarify the results for the readers, the RNA expression fold changes should be similarly calculated and shown in each part Figure 5. Also all fonts should be similar in this figure and throughout the manuscript.

In the Figure 2, the loading of the samples is uneven. The use of M14 melanoma cells is not discussed in the manuscript

Reviewer #2, expert in prostate cancer (Remarks to the Author):

The major claims of this paper relate to the effect of activating the AR present within infiltrating macrophages in supporting the invasive potential of prostate cancer cells. This study builds on recent work of the same group which seeks to understand the often-overlooked significance of AR located within the stroma of the prostate gland. In essence, this study together with their recent study focused on fibroblasts, seeks to address the totality of androgen signalling effects within the tumour microenvironment.

The findings provide a major advance in our understanding of how intra-prostatic androgen synthesis may affect the immune infiltrate and support localised invasion and perhaps systemic dissemination. The major limitation of this work is that the experimental approach focuses exclusively on localised invasion and does not extend to understand the significance to metastatic spread to lymph nodes or dissemination through the circulation.

The experimental approach is appropriate and the work would appear to have been executed to a high level of technical competence. The statistical analysis is appropriate. The genomic data is clearly presented. The use of both THP-1 and primary macrophage isolates provides some measure of confidence in the validity and reproducibility of the data. All of the invasion responses are clear and without ambiguity. The pathology and IF data on tumour samples is of sufficient quality to permit interpretation.

The conclusions drawn are appropriate and are supported by the data. Perhaps the only area of contention relates to the inability of TREM-1 knockdown to reduce CXCL8 expression - the p value is quoted as 0.06 and while all of the other cytokines are adequately repressed to baseline, this is not observed for CXCL8. Therefore, is this simply a matter of some experimental variability in the assay for this chemokine and thus requires more replicates or is this indicative of the CXCL8 not being dependent on TREM-1 signalling for testosterone-induced expression - are there other potential interacting TFs that could be shown to be induced by R1881 that could explain a more direct effect on CXCL8 gene expression. Alternatively, could this relate to differential mRNA stability and thus if repeated at a different time point, the true effect in reversing CXCL8 gene expression could be observed.

The limitation of the paper is that the phenotypic response (ie invasion) is only demonstrated in the context of a single assay, using co-culture and addition of conditioned media from treated cells. The capacity of these macrophages to either underpin and support androgen-promoted systemic dissemination using more advanced experimental approaches (e.g the chick CAM assay) may increase the impact and indeed clinical significance of this study.

Reviewer #3, expert in macrophage biology (Remarks to the Author):

The present manuscript entitled: Androgen Receptor Signaling in Macrophages Promotes TAM Polarization and TREM-1 mediated Prostate Cancer Cell Migration and Invasion by Andries Bergman and co-workers, complements a previous report of the same group concluding that loss of androgen receptor signaling in prostate cancer-associated fibroblasts promotes CCL2- and CXCL8-mediated cancer cell migration.

In the present manuscript Bianca Cioni demonstrate that the human monocytic cell line THP-1; monocyte derived macrophages, and possibly also tumor-associated macrophages (TAM) in prostate cancer (PCa) express, and are regulated by, androgen receptor (AR): First they claim to have identified CD163 and AR co-expressing cells in biopsies of PCa patients (based on immunofluorescence staining of tissue sections). The authors subsequently demonstrate that activated THP-1 cells and monocyte derived macrophages can be activated / stimulated to express AR (although at a lower level than the PCa cell lines CWR-R1, LNCaP). Upon AR agonist- specific activation, differentiated THP-1 cells secrete chemotactically active mediators, which can recruit PCa cell lines in a scratch assay in vitro.

The TREM1-signaling pathway was subsequently identified as a prominent AR signaling -regulated pathway. In differentiated THP1 cells, TREM1 mRNA was up-regulated twofold, and TREM1-induced genes, notably, CCL2 CCL13, and CCL7 were found to be expressed also at higher levels in the presence of the AR agonist R1881 (intriguingly, also the immunoregulatory cytokine IL10 becomes up-regulated by R1881 treatment; this finding (Fig. 7; Suppl. Figure S5) however, was not further discussed).

Main suggestions

In its present form, the manuscript represents quite an impressive analysis on how AR is induced in THP-1 cells and how AR activation (by administration of the agonist, R1881) may affect the functional differentiation and activity of THP-1 cells (and of monocyte-derived macrophages), including an up-regulation of TREM1 and TREM-1 signaling. The link, however, between findings made with differentiated THP-1 cells, and monocyte derived macrophages cultured in vitro, and TAM's from PCa is at present largely indirect.

Accordingly, the main suggestions and open questions include:

(i) A closer and more detailed (re-) analysis of the scRNASeq data of TAM's (shown in part in Fig. 1A, 4F) will be instrumental to demonstrate a causal relationship of AR expression and signaling; induced TREM1 transcription, and enhanced TREM1-induced transcription of the chemotactic factors (or TREM1-induced other factors) in the same individual TAM, derived from PCa biopsies.

(ii) PCa cells: to corroborate the findings made with THP-1 cells and the chemotactic activity observed in the scratch test for prostate tumor cell lines in supernatant of conditioned THP-1 cells, it will be critical to define the chemokine receptor expression pattern of primary PCa cells obtained from biopsies of the primary tumor (or even from the site of metastasis) to confirm that the available chemotactic receptors would also allow be suitable for a chemotactic recruitment across a chemokine gradient generated by TAM's upon TREM1 activation as observed with the R1881-conditioned THP1 supernatant.

(iii) Data provided by the authors seems to demonstrate an up-regulation of TREM-1 by triggering AR

on activated THP-1 cells. They further claim that the enhanced expression of TREM1 increased the cytokine production by THP-1 cells. This assumption is based on a comparison of in vitro cultures in the presence, or absence, of the TREM1-derived antagonistic peptide, LP17. However, enhanced TREM1 expression alone is not sufficient for TREM1 signaling: here the presence of TREM1 ligands is obviously required; unless the TREM-1 derived peptide LP17 also exerts off-target effects, e.g. by preventing the binding of ligands to additional receptors distinct from TREM1: hence, in their system THP-1-cells (or contaminants in the cell culture media) are able to produce TREM1 ligand(s): Hence, appropriate controls need to be included (anti-TREM-1 activation with cross-linked, agonistic antibodies, use of TREM-1 neutralizing antibodies, TREM1 deficient macrophages (RNAi knock-down, CRISP-Cas) and THP-1 cells should be used to rule out TREM-1 - independent effects. In addition, identification of the TREM1 agonists in the supernatant of AR-agonist treated, THP-1 cells will greatly enhance the impact of this manuscript.

(iv) Is TREM1 equally induced in all TAM subsets; is it always associated with AR expression/signaling; and/or do also macrophages which were recently recruited to the tumor contribute to the increase frequency TREM1 expressing M ϕ 's in PCa? Most of this relevant information should be obtained by a careful re-analysis of the scRNASeq data of the TAM's.

(v) What are the consequences of TREM-1 cross-linking on TAM's obtained from in PCa biopsies, ideally in the presence or absence, of R1881): are the findings identical to those obtained with THP-1 cells of monocyte derived macrophages in vitro??

(vi) Last but not least: Provided all the findings reported in the present study in activated THP-1 cells and CWR-R1 cell lines are also operative in PCa-derived TAM's and prostate tumor cells: how will these findings affect the biological behavior of the primary, and / or metastatic prostate tumor cells: Provided, these mechanisms are operative in primary tumors, one might conclude that they are responsible for maintaining the cells within the primary tumor (based on the local gradient of chemotactic ligands, formed by the resident TAM's), rather than leaving the primary tumor via lymphatics or vasculature. A discussion of their findings would, thus, be most instructive and helpful, particularly, when backed up with additional data on the single cell RNA profiling of TAM's (and neighboring PCa cells). Such a discussion might also include a model that reconciles the previous findings of the same group in PCa associated fibroblasts where loss of androgen receptor signaling was found to promote CCL2- and CXCL8-mediated cancer cell migration

Minor suggestions

Figure 8

Given the generally low surface expression levels of CD163 on macrophages, it might be helpful to provide information on the gating strategy to define CD163+ vs CD163- macrophages (same as in Fig 7? (and also indicate the MFI's for both groups). Furthermore, it might be helpful to obtain information on whether the observed lower frequency of CD163+ macrophages in anti-androgen treated patients is indeed due to a reduction in the number of CD163 positive macrophages, rather than an increase in the number of CD163 negative macrophages.

Reviewers' comments:

Reviewer #1, expert in AR signalling:

In the manuscript entitled “Androgen Receptor Signaling in Macrophages Promotes TAM Polarization and TREM-1 mediated Prostate Cancer Cell Migration and Invasion”, Cioni and co-workers studied androgen receptor (AR) signalling in macrophages, especially whether AR activation regulates prostate cancer migration and invasion. To address these effects, the authors utilized THP-1 monocytic leukaemia cells, CWR-R1 and LNCaP prostate cancer cell lines as models for the studies in vitro and fractions of tumour- associated macrophages in the prostate cancer-associated stroma biopsied from patients treated with anti-androgen therapy.

The authors conclude that AR signalling in macrophages supports prostate cancer cell migration and invasion, and that inhibition of AR signalling in macrophages is a newly identified mechanism of action of anti-androgen therapy.

To be able to warrant the statements in the manuscript, major modifications and strengthening of the data are needed. The manuscript is adequately written, but modifications also to the figures are needed to clearly describe the results. The manuscript could be improved by considering the following issues:

Response:

We would like to thank the reviewer for the valuable suggestions and excellent comments. As the reviewer can see in the revised version of our manuscript, and addressed in the point-by-point letter below, we did our utmost best to assess all the issues that were raised.

Major concerns:

1) Authors conclude that the migration of CWR-R1 cells cultured in THP-1 conditioned medium affects CWR-R1 cell migration due to activated AR signalling in THP-1 cells. However, the authors state that “the genomic actions of AR in fibroblasts were recently described by us and others groups” (page 14, line 334). The CWR-R1 cell line is a co-culture of epithelial cells and stromal fibroblasts. How do the authors excluded that the effects seen on the migration in the THP-1-conditioned medium exposed CWR-R1 cells is not due to activated AR signalling in the fibroblasts of the CWR-R1 cell line?

Response:

We thank the reviewer for this constructive comment. In the present study we show that culture of CWR-R1 cells in conditioned medium of AR blocked THP-1 cells, reduces their migration and invasion ability. However, the reviewer raises an excellent point, that a mixed tumor/fibroblast nature of CWR-R1 cells would complicate interpretation of our results. To address this point, and to further strengthen our initial statement that AR blockade in macrophages inhibits prostate cancer cell migration and invasion, we now repeated the entire experiment using the prostate cancer cell line PC3.

PC3 cells are pure epithelial, AR negative and with that androgen independent. In PC3 cells we also observed inhibition of migration when cultured in conditioned medium from THP-1 cells that were exposed to AR inhibition, fully confirming our original observations.

Changes to the manuscript:

Supplementary Figure S3 and additional text was added on page 6, 7 and 42.

2) In the Figure 3B, the authors show the results for CWR-R1 cell line cultured with THP-1 conditioned medium. In the materials and methods section, they mention that “conditioned medium was then collected and used 1:4 with FBS-RPMI medium (normal medium) for proliferation, migration and invasion assays” (page 19, line 464). However, the difference between “Normal medium” and “Vehicle CM” is unclear. Is the effect on the cell migration between Normal medium and Vehicle CM seen in Figure 3B and C seen due to the fact that “Normal medium” contains 2.5% more FBS than the “Vehicle CM”? Are there migration differences seen between the “Normal medium” and the “Vehicle CM” conditions without pre-incubation with THP-1 cells? Does the DCC-RPMI alone affect the CWR-R1 cell migration? The medium conditions should be clearly defined in the manuscript.

Response:

We apologize for not clearly defining ‘normal medium’ as opposed to ‘vehicle CM’ in our Materials and Methods section. This is now corrected in the revised version of our manuscript.

‘Normal medium’ contains 10% FBS-RPMI without conditioned medium (CM) from THP-1 cells and it was included as a baseline control for the CWR-R1 migration experiments, and now added PC3 cells, in full medium. ‘Vehicle CM’, is a mixture of hormone-deprived THP-1 cells CM containing 5% charcoal-stripped FBS-RPMI and normal medium in a 1:4 ratio. Therefore, ‘Normal medium’ contains 1.0% more FBS than ‘vehicle CM’. Nonetheless, cell migration is increased in ‘vehicle CM’ as compared to ‘normal medium’, indicating that

conditioning the medium by macrophages increases migration rate, which could be fully blocked by exposing the macrophages to RD162. With that, we conclude that AR-regulated secreted factors in the macrophages are responsible for the observed impact on migration.

Changes to the manuscript:

Description of the various media is now further clarified on page 26 and 27. As suggested by the reviewer, we now added the migration speed of CWR-R1 cells cultured in DCC-RPMI only (Figure 3B and C).

3) In the figure 6 panels A and B, the Vehicle CM effect on migration of CWR-R1 cells is shown after 3 days. However, the wound closure rate relative to day 0 is totally different from the result seen in the Figure 3 where the wound was almost closed already after one day. In the Figure 6, the closure rate relative to day 1 is 2.0 with vehicle CM at the time of 3 days, whereas in the figure 3B the closure rate under the same conditions is almost 6.0. How was the closure rate measured? How many times were these experiments (Figure 3 and 6) repeated?

Response:

The reviewer raises an excellent point, and is completely right to state that the 'closure rate relative to day 0' of the scratch in the CWR-R1 culture in Vehicle CM is different between Figure 3C and Figure 7C (5.8 and 2.1, respectively), while cells in Figure 3C were cultured for 1 day and in Figure 7C for 3 days. This can be explained by differences between the conditioned medium used for the experiments. The Macrophage density and with that the concentration of cytokines in the conditioned medium may vary between batches with variation in migration rate of PCa cells between experiments as a result. The experiments are continued until the scratch in one of the conditions is almost closed and depending on the batch of conditioned medium this might vary in duration. Moreover, differences in migration were also established in trans-well assays to confirm the findings in the scratch assays. The experiments presented in Figure 3 was performed three times (three technical replicates), while experiments presented in Figure 6 (now Figure 7) were performed twice (three technical replicates).

Changes to the manuscript:

A comment on the variability of the duration of the scratch assay is placed on page 27. Description of the experiments is now clarified on page 27, 35, 39 and 42.

4) The amount of biological and technical replicates for the assays nor the star abbreviations

for calculations of significance are not clarified throughout the manuscript.

Response:

These omissions have been corrected, and we thank the reviewer for highlighting this.

Changes to the manuscript:

The number of biological and technical replicates of the various assays have now been added to the legends, as well as the levels of statistical significance (page 35, 37-40, 43, 44, 46).

5) Authors stimulated THP-1 cells with PMA and then used LPS and IFN- γ for TAM polarization. LPS and IFN- γ stimulation of M Φ s polarizes M Φ s to M1 type macrophages which are generally thought to prevent the cancer cell locomotion and tumour development. In this study, the authors discuss the role of AR in M Φ s in relation to human prostate cancer progression. How are the results of this manuscript related to the findings in the field? Would M Φ polarization to type M2 macrophages have different effect?

Response:

The reviewer raises an interesting point. In our study, THP-1 cells were stimulated only with PMA into M0 macrophages or with PMA, LPS and IFN- γ into M1 macrophages. We showed that PMA, LPS and IFN- γ stimulated THP-1 cells have an M1 macrophage like phenotype, based on low expression of the M2 markers CD163 and CD206. AR stimulation resulted in higher expression of these markers suggesting a phenotypic switch into M2 macrophages (Figure 8C). This is further supported by the increased expression of IL10 and CD209 upon AR stimulation, which are also M2 markers (Figure 8B). The results presented in Figure 8, suggest that this phenotypic switch also occurs in human prostate cancer. The increased migration, invasion and growth of prostate cancer cells upon culturing in AR stimulated, hence M2 polarized, macrophages medium fits with the commonly accepted hypothesis that M2 macrophages stimulate prostate cancer development.

Changes to the manuscript:

None

Minor issues:

1) It is very difficult to compare the RNA expression fold changes in the Figure 5. To clarify the results for the readers, the RNA expression fold changes should be similarly calculated

and shown in each part Figure 5. Also, all fonts should be similar in this figure and throughout the manuscript.

Response:

We fully agree with this reviewer

Changes to the manuscript:

All figures on RNA expression are now 'fold-change from base line" (Figure 6). Font sizes are now similar in all figures.

2) In the Figure 2, the loading of the samples is uneven. The use of M14 melanoma cells is not discussed in the manuscript

Response:

AR expression is low in THP-1 cells as compared to human prostate cancer LNCaP cells. In case of equal loading of both, AR in THP-1 would be invisible or AR in LNCaP overexposed. Therefore, this figure is selected to show the large difference in AR expression levels between THP-1 and LNCaP cells.

Changes to the manuscript:

The difference in loading is now mentioned in the legend of figure 2 (page 34). The absence of AR expression in M14 human melanoma cells is now described on page 5,25 and 34.

Reviewer #2, expert in prostate cancer:

The major claims of this paper relate to the effect of activating the AR present within infiltrating macrophages in supporting the invasive potential of prostate cancer cells. This study builds on recent work of the same group which seeks to understand the often-overlooked significance of AR located within the stroma of the prostate gland. In essence, this study together with their recent study focused on fibroblasts, seeks to address the totality of androgen signalling effects within the tumour microenvironment.

The findings provide a major advance in our understanding of how intra-prostatic androgen synthesis may affect the immune infiltrate and support localised invasion and perhaps systemic dissemination. The major limitation of this work is that the experimental approach

focuses exclusively on localised invasion and does not extend to understand the significance to metastatic spread to lymph nodes or dissemination through the circulation.

The experimental approach is appropriate and the work would appear to have been executed to a high level of technical competence. The statistical analysis is appropriate. The genomic data is clearly presented. The use of both THP-1 and primary macrophage isolates provides some measure of confidence in the validity and reproducibility of the data. All of the invasion responses are clear and without ambiguity. The pathology and IF data on tumour samples is of sufficient quality to permit interpretation.

We are delighted to see the reviewer finds our work a major advance in our understanding, and considers the work executed to a high level of technical competence. We thank the reviewer for the valuable comments and suggestions, which we addressed point-by-point below.

1) The conclusions drawn are appropriate and are supported by the data. Perhaps the only area of contention relates to the inability of TREM-1 knockdown to reduce CXCL8 expression - the p value is quoted as 0.06 and while all of the other cytokines are adequately repressed to baseline, this is not observed for CXCL8. Therefore, is this simply a matter of some experimental variability in the assay for this chemokine and thus requires more replicates or is this indicative of the CXCL8 not being dependent on TREM-1 signalling for testosterone-induced expression - are there other potential interacting TFs that could be shown to be induced by R1881 that could explain a more direct effect on CXCL8 gene expression. Alternatively, could this relate to differential mRNA stability and thus if repeated at a different time point, the true effect in reversing CXCL8 gene expression could be observed.

Response:

CXCL8 is indeed the only TREM-1 related cytokine where reduction of expression by inhibition of TREM-1 does just not reach the threshold of significance (0.06). The trend in changes of expression in the various conditions is however, comparable with the other TREM-1 related cytokines. We do find AR binding at the loci of various TREM-1 associated cytokines (Supplementary Figure S5), including CXCL8. Nevertheless, other transcription factors may also control CXCL8 expression, therefore, we cannot rule out that CXCL8 expression is also controlled by TREM-1 independent regulation. However, our results do suggest that TREM-1 is a major regulator of CXCL8 expression.

Changes to the manuscript:

We have now mentioned the above in the discussion section (page 19).

2) The limitation of the paper is that the phenotypic response (ie invasion) is only demonstrated in the context of a single assay, using co-culture and addition of conditioned media from treated cells. The capacity of these macrophages to either underpin and support androgen-promoted systemic dissemination using more advanced experimental approaches (e.g the chick CAM assay) may increase the impact and indeed clinical significance of this study.

Response:

We thank the reviewer for this extremely helpful and valuable suggestion. As suggested by the reviewer, we performed a chick CAM assay, grafted with human prostate cancer PC3 or LNCaP cells. Cells were exposed to vehicle and RD162 exposed macrophage conditioned medium, CCL2 and NaCl. Tumours grew biggest in CCL2 and vehicle CM conditions, while macrophage exposed to RD162 conditioned medium inhibited growth compared to NaCl. As we didn't observe impact of macrophage-conditioned medium in 2D cell culture experiments (Figure S2), we conclude that macrophages excrete soluble factors that are under control of AR, that also stimulate anchorage independent growth of prostate cancer cells. The data are further strengthened by the use of both AR-negative PC3 cells, which is completely in agreement with the observations made in AR+ LNCaPs. Results are described in the Results section (page 7,8). Photographs of the PC3 CAM assays have been added (Figure 4), while LNCaP results are shown in Supplementary Figure S4). To assess Epithelial Mesenchymal Transition (EMT) and migration of PC3 cells, expression of human vimentin was established in the chorioallantoic membrane of the CAM assay. Human vimentin showed a trend towards lower expression in RD162 CM conditions than in vehicle CM conditions, suggesting increased EMT and migration of PC3 cells by AR stimulation of macrophages.

Changes to the manuscript:

These data have been added in the manuscript (Figure 4 and Supplementary Figure S4, page 7,8 of the Results section, page 28 of Material and Method and page 36 and 43 of the Figure Legends).

Reviewer #3, expert in macrophage biology:

The present manuscript entitled: Androgen Receptor Signalling in Macrophages Promotes TAM Polarization and TREM-1 mediated Prostate Cancer Cell Migration and Invasion by Andries Bergman and co-workers, complements a previous report of the same group concluding that loss of androgen receptor signalling in prostate cancer-associated fibroblasts promotes CCL2- and CXCL8-mediated cancer cell migration.

In the present manuscript Bianca Cioni demonstrate that the human monocytic cell line THP-1; monocyte derived macrophages, and possibly also tumor-associated macrophages (TAM) in prostate cancer (PCa) express, and are regulated by, androgen receptor (AR): First they claim to have identified CD163 and AR co-expressing cells in biopsies of PCa patients (based on immunofluorescence staining of tissue sections). The authors subsequently demonstrate that activated THP-1 cells and monocyte derived macrophages can be activated / stimulated to express AR (although at a lower level than the PCa cell lines CWR-R1, LNCaP). Upon AR agonist- specific activation, differentiated THP-1 cells secrete chemotactically active mediators, which can recruit PCa cell lines in a scratch assay in vitro. The TREM1-signaling pathway was subsequently identified as a prominent AR signaling - regulated pathway. In differentiated THP1 cells, TREM1 mRNA was up-regulated twofold, and TREM1-induced genes, notably, CCL2 CCL13, and CCL7 were found to be expressed also at higher levels in the presence of the AR agonist R1881 (intriguingly, also the immunoregulatory cytokine IL10 becomes up-regulated by R1881 treatment; this finding (Fig. 7; Suppl. Figure S5) however, was not further discussed).

We thank the reviewer for the valuable suggestions and recommendations, which clearly helped us to further improve the quality of our manuscript. We did our utmost best to address all questions raised by the reviewer, which are all discussed point-by-point further below.

Main suggestions

In its present form, the manuscript represents quite an impressive analysis on how AR is induced in THP-1 cells and how AR activation (by administration of the agonist, R1881) may affect the functional differentiation and activity of THP-1 cells (and of monocyte-derived macrophages), including an up-regulation of TREM1 and TREM-1 signaling. The link, however, between findings made with differentiated THP-1 cells, and monocyte derived

macrophages cultured *in vitro*, and TAM's from PCa is at present largely indirect. Accordingly, the main suggestions and open questions include:

(i) A closer and more detailed (re-) analysis of the scRNASeq data of TAM's (shown in part in Fig. 1A, 4F) will be instrumental to demonstrate a causal relationship of AR expression and signaling; induced TREM1 transcription, and enhanced TREM1-induced transcription of the chemotactic factors (or TREM1-induced other factors) in the same individual TAM, derived from PCa biopsies.

Response:

We thank this reviewer for the valuable suggestions. We fully agree with the reviewer that an association between AR signalling, TREM-1 transcription and expression of chemotactic factors in human prostate cancer derived macrophages would make a strong case for a causal and clinically relevant relationship. However, we would like to indicate that biopsies were taken from testosterone proficient patients as is the current state-of-the-art in prostate cancer clinical care, and none were treated with an AR signalling inhibiting drug prior to surgery. Therefore, scRNASeq in CD14+ and CD11b+ cells only allowed us to evaluate expression of the genes at a set testosterone level (namely the patients testosterone level). As the design of the study was aimed to adhere to the physiological reality as close as possible, isolated macrophages that were submitted for scRNASeq were not exposed to AR stimulating or inhibiting drugs *in vitro*. Therefore, only AR expression, but not AR signalling could be assessed in relation to TREM-1 and associated cytokine expression. Furthermore, assessments of possible relations between expression of AR, TREM-1 and cytokines is hampered by the limited heterogeneity of AR expression among HLA-DR and HLA-DP high CD14+ and CD11b+ cells (Figure 1A).

Nevertheless, as requested by the reviewer, we included expression levels of various genes in Supplementary Figure S14, including the 'general' macrophages markers CD68 and CSFR1, TREM1 and the TREM-1 associated cytokines CXCL8, CCL2, CCL3, CCL13 and CCL7 (panel A), the M2 macrophage markers CD206, CD163, IL-10 and CD209 (panel B) and the M1 markers STAT1, IL-12, CD80 and CXCL10 (panel C). AR expressing CD14+ and CD11b+ cells (Figure 1A) express the "general" macrophage markers CD68 and CSFR1. TREM-1 and its associated cytokines CXCL8, CCL3 and CCL7 are expressed by a subpopulation of AR expressing cells. A correlation plot of TREM-1 and the prominent cytokine CCL2 is added as Figure S14D. As expected in macrophages taken from the tumorous side of the prostate, M2 markers CD206, CD163 and IL-10 were present in AR

expressing CD14+ and CD11b+ cells, while all M1 markers were expressed at very low levels, suggesting that the native macrophages were M2 polarized.

Changes to the manuscript:

Figure S14 was added, showing the tsne plots of expression of the markers mentioned above in CD14+ and CD11b+ cells. New text describing the tsne plots was added on page 14 of the Results section, and page 46 of the Figure Legends section.

In order to improve the presentation of the potential clusters of cells based on gene expression analysis, the tSNE plots in Figure 1A, Figure 5F and Supplementary Figure S14, all axes are now shown in log scale.

(ii) PCa cells: to corroborate the findings made with THP-1 cells and the chemotactic activity observed in the scratch test for prostate tumor cell lines in supernatant of conditioned THP-1 cells, it will be critical to define the chemokine receptor expression pattern of primary PCa cells obtained from biopsies of the primary tumor (or even from the site of metastasis) to confirm that the available chemotactic receptors would also allow be suitable for a chemotactic recruitment across a chemokine gradient generated by TAM's upon TREM1 activation as observed with the R1881-conditioned THP1 supernatant.

Response:

The reviewer raises a very relevant issue. Multiple cytokine receptors (CCRs) have been described, while there is little known about their specificity for specific cytokines. There are few receptors that bind a single ligand, and several chemokines can bind to more than one receptor (Proudfoot 2002, Zlotnik and Yoshie 2000, Bennett, Fox and Signoret 2011). As suggested by the reviewer, a Tissue Micro Array (TMA) containing primary prostate cancer and a TMA containing pelvic lymph node metastases were stained for the chemokines receptors CCR1-5. We found CCR3 and CCR4 expression in the prostate cancer cells in primary cancer and in the lymph node metastases, while expression was absent in normal prostate epithelial cells. Staining was highest in the infiltrated immune cells in the stroma compartment. These data suggest that CCR3 and CCR4 may be the driving factors in the observed pro-metastatic phenotype.

Changes to the manuscript:

Supplementary Figure S12 and new text was added on page 13 of the Results, page 19 of the Discussion section, page 23 of the Material and Methods section and page 46 of the Figure Legends section.

(iii) Data provided by the authors seems to demonstrate an up-regulation of TREM-1 by triggering AR on activated THP-1 cells. They further claim that the enhanced expression of TREM1 increased the cytokine production by THP-1 cells. This assumption is based on a comparison of in vitro cultures in the presence, or absence, of the TREM1-derived antagonistic peptide, LP17. However, enhanced TREM1 expression alone is not sufficient for TREM1 signalling: here the presence of TREM1 ligands is obviously required; unless the TREM-1 derived peptide LP17 also exerts off-target effects, e.g. by preventing the binding of ligands to additional receptors distinct from TREM1: hence, in their system THP-1-cells (or contaminants in the cell culture media) are able to produce TREM1 ligand(s): Hence, appropriate controls need to be included (anti-TREM-1 activation with cross-linked, agonistic antibodies, use of TREM-1 neutralizing antibodies, TREM1 deficient macrophages (RNAi knock-down, CRISP-Cas) and THP-1 cells should be used to rule out TREM-1 - independent effects. In addition, identification of the TREM1 agonists in the supernatant of AR-agonist treated, THP-1 cells will greatly enhance the impact of this manuscript.

Response:

Indeed, the reviewer has a valid point. The synthetic polypeptide inhibitor of TREM-1 signaling, LP17, mimics short highly interspecies conserved domains of TREM-1 that reduces the cytokine production of monocytes and macrophages (Gibot et al. 2008). LP17 competes with TREM-1 ligands for binding to TREM-1 (Pelham, Pandya and Agrawal 2014). Multiple studies have shown that this peptide is efficient in not only preventing but also in blocking the deleterious effects of pro-inflammatory cytokines (Tammaro et al. 2016, Weiss et al. 2017, Klesney-Tait, Turnbull and Colonna 2006, Schenk et al. 2007, Varanat et al. 2017, Gibot et al. 2006). However, as the reviewer states, LP17 may also has off-target effects. Therefore, other means to manipulate TREM-1 and study its downstream effects would be of added value to further support and confirm our findings. However, here we face technical limitations that could not be resolved in the model systems that were available to us.

Measuring levels of the TREM-1 ligand would definitely help, however, there is no consensus on what the ligand is (Kozik et al. 2016, Haselmayer et al. 2007, Zanzinger et al. 2009, Wu et al. 2012, Read et al. 2015). As advised, we made several attempts to make

knocked down TREM-1 in THP-1 monocytes using shRNA, followed by LPS priming which is essential for TREM-1 activation (Carrasco et al. 2018). The THP-1 cells were very intolerant to transfection with both shTREM-1 and sh-scrambled and subsequent LPS activation, leaving very few cells viable to assess. One 2007 paper described knock down of TREM-1 in THP-1 cells using short-hairpins, which was never repeated or confirmed by others, to our knowledge (Ornatowska et al. 2007). An alternative approach to confirm our TREM1 blocking peptide data, would be the use of an inhibitory antibody. However, results from the commercially available TREM-1 blocking antibodies have not been published nor is information provided on the type of antibody. This makes the added value of results obtained with these antibodies highly questionable.

At least six high-impact papers have used LP17 as an TREM-1 inhibitor (Tammaro et al. 2016, Weiss et al. 2017, Klesney-Tait et al. 2006, Schenk et al. 2007, Varanat et al. 2017, Gibot et al. 2006). The reason why LP17 is the most frequently applied modulator of TREM-1, is most likely explained by the above. Consequently, even though the reviewer raises a valid point, technical limitations prevent us to fully address this issue.

Changes to the manuscript:

We have now incorporated this in the discussion section, in which we indicate that other signalling cascades may be affected by LP17 as well, as technical restrictions (as mentioned above) prevent us to fully proof this (page 19).

(iv) Is TREM1 equally induced in all TAM subsets; is it always associated with AR expression/signaling; and/or do also macrophages which were recently recruited to the tumor contribute to the increase frequency TREM1 expressing Mø's in PCa? Most of this relevant information should be obtained by a careful re-analysis of the scRNASeq data of the TAM's.

Response:

TREM-1 is a marker of macrophage activation, thus expression in macrophages is increased during the inflammatory reaction.

The question if TREM-1 expression is associated with AR signaling or expression in macrophages, cannot be adequately addressed in our data set. As lined out in our response to question (i) from this reviewer, we only explored expression of genes in native macrophages without manipulation of AR signaling. However, as shown in the new Supplementary Figure S14, TREM-1 is expressed in all AR expressing macrophages, but at different levels. TREM-1 was strongly expressed in the population of macrophages expressing high levels of the chemokines CXCL8, CCL3 and CCL7 and TAMs markers IL-

10, CD206 and CD163. However, very low expression was observed for the M1 markers CD80 and STAT1, so that no correlation with TREM1 levels could be studied. The low expression of M1 markers suggests limited representation of this cell type in the biopsies, which might be related to AR signalling in macrophages at the cancerous site of the prostate where the biopsies were taken. It is likely, that the population macrophages submitted for single cell RNA sequencing were at various stages of differentiation into M2, which explains the differences in expression of markers, TREM-1 and cytokines. These results suggest, that TREM-1 expression is increased upon M2 differentiation, as is supported by our *in vitro* data.

Changes to the manuscript:

Additional text on this subject can be found on page 14 of the Results section.

(v) What are the consequences of TREM-1 cross-linking on TAM's obtained from in PCa biopsies, ideally in the presence or absence, of R1881): are the findings identical to those obtained with THP-1 cells of monocyte derived macrophages in vitro??

Response:

This reviewer made a very relevant comment. However, the average number of macrophages we could isolate from prostate cancer biopsies is very low which prevents the execution of additional functional experiments. For this reason, we decided to implement monocyte-derived macrophages and macrophages cell line models to study AR signalling in macrophages.

(vi) Last but not least: Provided all the findings reported in the present study in activated THP-1 cells and CWR-R1 cell lines are also operative in PCa-derived TAM's and prostate tumor cells: how will these findings affect the biological behavior of the primary, and / or metastatic prostate tumor cells: Provided, these mechanisms are operative in primary tumors, one might conclude that they are responsible for maintaining the cells within the primary tumor (based on the local gradient of chemotactic ligands, formed by the resident TAM's), rather than leaving the primary tumor via lymphatics or vasculature. A discussion of their findings would, thus, be most instructive and helpful, particularly, when backed up with additional data on the single cell RNA profiling of TAM's (and neighboring PCa cells). Such a discussion might also include a model that reconciles the previous findings of the same group in PCa associated fibroblasts where loss of androgen receptor signaling was found to promote CCL2- and CXCL8-mediated cancer cell migration.

Response:

We thank the reviewer for this comment and we updated the discussion section with the suggested observations. Sadly, additional scRNA-seq data on neighbouring PCa cells is not feasible anymore, as these samples were prospectively collected after which the macrophages were isolated for scRNA-seq. As this procedure requires freshly collected specimens, it would not be possible to retrospectively collect and analyse PCa for these samples for scRNA-seq. In addition, albeit highly interesting, we respectfully consider such an analysis beyond the scope of the current work.

Changes to the manuscript:

New text can be found on page 19 of the Discussion section.

Minor suggestions

Figure 8

Given the generally low surface expression levels of CD163 on macrophages, it might be helpful to provide information on the gating strategy to define CD163+ vs CD163- macrophages (same as in Fig 7? (and also indicate the MFI's for both groups). Furthermore, it might be helpful to obtain information on whether the observed lower frequency of CD163+ macrophages in anti-androgen treated patients is indeed due to a reduction in the number of CD163 positive macrophages, rather than an increase in the number of CD163 negative

Response:

CD163+ cells of Figure 7, now Figure 8, were defined based on intensity of staining within the cytoplasm. This threshold was tested on multiple samples and then re-checked by the pathologist. The pathologist's assessment was subsequently used to refine the cut-offs for gating, when required. Absolute number of CD163- and CD163+ cells is now added in Table S5. Importantly, as reported in the Supplementary Table S5, no significant differences in the CD163- population were observed in the two groups of patients, suggesting that the observed reduced percentage of CD163+ cells in treated patients was mostly mediated by reduced TAMs polarization.

Changes to the manuscript:

Supplementary Figure S15 was added, showing the identification of true CD163 positive and false CD163 positive cells in prostate cancer biopsies. Additional text can be found on page 15-16 of the Results section and page 47 of the Figure Legends section.

Furthermore, as requested by the reviewer, we now provide details on the gating strategy of the flow cytometry analysis of MDM cells upon stimulation with testosterone (Supplementary Figure S13, page 14 of the Results section and page 46 of the Figure Legends section).

References

- Bennett, L. D., J. M. Fox & N. Signoret (2011) Mechanisms regulating chemokine receptor activity. *Immunology*, 134, 246-56.
- Carrasco, K., A. Boufenzler, L. Jolly, H. Le Cordier, G. Wang, A. J. Heck, A. Cerwenka, E. Vinolo, A. Nazabal, A. Kriznik, P. Launay, S. Gibot & M. Derive (2018) TREM-1 multimerization is essential for its activation on monocytes and neutrophils. *Cell Mol Immunol*.
- Gibot, S., C. Buonsanti, F. Massin, M. Romano, M. N. Kolopp-Sarda, F. Benigni, G. C. Faure, M. C. Bene, P. Panina-Bordignon, N. Passini & B. Levy (2006) Modulation of the triggering receptor expressed on the myeloid cell type 1 pathway in murine septic shock. *Infect Immun*, 74, 2823-30.
- Gibot, S., F. Massin, C. Alauzet, C. Montemont, A. Lozniewski, P. E. Bollaert & B. Levy (2008) Effects of the TREM-1 pathway modulation during mesenteric ischemia-reperfusion in rats. *Crit Care Med*, 36, 504-10.
- Haselmayer, P., L. Grosse-Hovest, P. von Landenberg, H. Schild & M. P. Radsak (2007) TREM-1 ligand expression on platelets enhances neutrophil activation. *Blood*, 110, 1029-35.
- Klesney-Tait, J., I. R. Turnbull & M. Colonna (2006) The TREM receptor family and signal integration. *Nat Immunol*, 7, 1266-73.
- Kozik, J. H., T. Trautmann, A. Carambia, M. Preti, M. Lutgehetmann, T. Krech, C. Wiegard, J. Heeren & J. Herkel (2016) Attenuated viral hepatitis in Trem1^{-/-} mice is associated with reduced inflammatory activity of neutrophils. *Sci Rep*, 6, 28556.
- Ornatowska, M., A. C. Azim, X. Wang, J. W. Christman, L. Xiao, M. Joo & R. T. Sadikot (2007) Functional genomics of silencing TREM-1 on TLR4 signaling in macrophages. *Am J Physiol Lung Cell Mol Physiol*, 293, L1377-84.
- Pelham, C. J., A. N. Pandya & D. K. Agrawal (2014) Triggering receptor expressed on myeloid cells receptor family modulators: a patent review. *Expert Opin Ther Pat*, 24, 1383-95.
- Proudfoot, A. E. (2002) Chemokine receptors: multifaceted therapeutic targets. *Nat Rev Immunol*, 2, 106-15.
- Read, C. B., J. L. Kuijper, S. A. Hjorth, M. D. Heipel, X. Tang, A. J. Fleetwood, J. L. Dantzer, S. N. Grell, J. Kastrup, C. Wang, C. S. Brandt, A. J. Hansen, N. R. Wagtmann, W. Xu & V. W. Stennicke (2015) Cutting Edge: identification of neutrophil PGLYRP1 as a ligand for TREM-1. *J Immunol*, 194, 1417-21.

- Schenk, M., A. Bouchon, F. Seibold & C. Mueller (2007) TREM-1--expressing intestinal macrophages crucially amplify chronic inflammation in experimental colitis and inflammatory bowel diseases. *J Clin Invest*, 117, 3097-106.
- Tammaro, A., J. Kers, D. Emal, I. Stroo, G. J. D. Teske, L. M. Butter, N. Claessen, J. Damman, M. Derive, G. J. Navis, S. Florquin, J. C. Leemans & M. C. Dessing (2016) Effect of TREM-1 blockade and single nucleotide variants in experimental renal injury and kidney transplantation. *Sci Rep*, 6, 38275.
- Varanat, M., E. M. Haase, J. G. Kay & F. A. Scannapieco (2017) Activation of the TREM-1 pathway in human monocytes by periodontal pathogens and oral commensal bacteria. *Mol Oral Microbiol*, 32, 275-287.
- Weiss, G., C. Lai, M. E. Fife, A. M. Grabiec, B. Tildy, R. J. Snelgrove, G. Xin, C. M. Lloyd & T. Hussell (2017) Reversal of TREM-1 ectodomain shedding and improved bacterial clearance by intranasal metalloproteinase inhibitors. *Mucosal Immunol*, 10, 1021-1030.
- Wu, J., J. Li, R. Salcedo, N. F. Mivechi, G. Trinchieri & A. Horuzsko (2012) The proinflammatory myeloid cell receptor TREM-1 controls Kupffer cell activation and development of hepatocellular carcinoma. *Cancer Res*, 72, 3977-86.
- Zanzinger, K., C. Schellack, N. Nausch & A. Cerwenka (2009) Regulation of triggering receptor expressed on myeloid cells 1 expression on mouse inflammatory monocytes. *Immunology*, 128, 185-95.
- Zlotnik, A. & O. Yoshie (2000) Chemokines: a new classification system and their role in immunity. *Immunity*, 12, 121-7.

REVIEWERS' COMMENTS:

Reviewer #1 (Remarks to the Author):

The authors have adequately addressed my previous concerns and criticism.

Reviewer #2 (Remarks to the Author):

I am extremely satisfied with the extra experiments and explanations that have been added to the revised and resubmitted manuscript. The evidence collected from the additional experiments adds greater clarity and more significant understanding of how AR signalling within the macrophage population may potentiate hallmarks of tumour proliferation and dissemination.

Reviewer #3 (Remarks to the Author):

The authors carefully revised their manuscript. The issues raised in my first review are no experimentally addressed or discussed in a sufficient way.

One minor suggestion:

Re- off-target effects of LP17

Based on available literature, THP-1 cells are amenable to CRISPR-Cas9 mediated genomic editing, hence, this might be an option to generate TREM-1 deficient THP-1 cells to determine possible off-target effects of the administered LP17. However, this may be indeed considered too much of an effort. Accordingly, I consider the provided discussion of this issue in the revised manuscript appropriate. One may, however, wish to delete the introduced new sentence in the revised ms: (line 464/465): *Alternative means to mediate TREM-1 signaling are hampered by technical limitations* as this sentence is misleading: off-target effects of LP17 might include binding of this TREM-1 derived peptide to TREM-1 ligands, which may result in a different ligand conformation with altered affinity for other receptor(s), distinct from TREM-1.

Christoph Mueller

POINT-BY-POINT RESPONSE TO THE REVIEWERS COMMENTS

Manuscript, entitled "Androgen Receptor Signalling in Macrophages Promotes TREM-1 mediated Prostate Cancer Cell Migration and Invasion" by Cioni, Zaalberg and colleagues (NCOMMS-18-24723A)

Reviewer #1 (Remarks to the Author):

The authors have adequately addressed my previous concerns and criticism.

Response:

We thank this reviewer for his/her second assessment of our manuscript and are happy that all concerns were adequately addressed.

Reviewer #2 (Remarks to the Author):

I am extremely satisfied with the extra experiments and explanations that have been added to the revised and resubmitted manuscript. The evidence collected from the additional experiments adds greater clarity and more significant understanding of how AR signalling within the macrophage population may potentiate hallmarks of tumour proliferation and dissemination.

Response:

We thank this reviewer for assessing our resubmitted manuscript and for judging the new manuscript as clearer and of higher significance.

Reviewer #3 (Remarks to the Author):

The authors carefully revised their manuscript. The issues raised in my first review are now experimentally addressed or discussed in a sufficient way.

One minor suggestion:

Re- off-target effects of LP17

Based on available literature, THP-1 cells are amenable to CRISPR-Cas9 mediated genomic editing, hence, this might be an option to generate TREM-1 deficient THP-1 cells to determine possible off-target effects of the administered LP17. However, this may be indeed considered too much of an effort. Accordingly, I consider the provided discussion of this issue in the revised manuscript appropriate.

'One may, however, wish to delete the introduced new sentence in the revised ms: (line 464/465): "Alternative means to mediate TREM-1 signaling are hampered by technical limitations" as this sentence is misleading':

off-target effects of LP17 might include binding of this TREM-1 derived peptide to TREM-1 ligands, which may result in a different ligand conformation with altered affinity for other receptor(s), distinct from TREM-1.

Response:

We thank this reviewer for reassessment of our manuscript. The sentence mentioned above is now deleted from the manuscript (Page 19).